# COGNITION-SUPERVISED LEARNING: CONTRASTING EEG SIGNALS AND VISUAL STIMULI FOR SALIENCY DETECTION

## ABSTRACT

In the rapidly evolving landscape of machine learning, the quest for efficient and accurate supervision signals remains paramount. Suitable supervision signals can be costly and in certain scenarios ineffective to obtain for models that require subjective cognitive labels, such as individual-specific interpretation of images or subjective training input for generative models. In this paper, we introduce a novel approach: *cognition-supervised learning*, leveraging human brain signals as direct supervisory signals. Using electroencephalogram (EEG) data, we contrastively train models to detect visual saliency without the need for any manual annotations. Our approach , the first of its kind, demonstrates that representations of semantic visual saliencies can be learned directly from EEG data. In downstream tasks, such as classification, clustering, and image generation, our learned representations not only reflect semantic saliency but also achieve competitive performance compared to models trained with manually labeled datasets. This work provides a promising avenue for future research in utilizing signals measured from the human cognitive system for supervising computer vision and machine learning models.

## 1 INTRODUCTION

Human cognition excels at detecting salient information, rapidly identifying what is important for an individual in a specific context or task. This innate ability to discern relevance is crucial for applications ranging from personalized content recommendations to user-centric interface designs. However, replicating this capability in machines, especially in a personalized manner that reflects an individual's intention, remains a significant challenge. Traditional machine learning approaches have often relied on large datasets of implicitly obtained signals, such as click data (Joachims et al., 2005; McAuley, in press; Shen et al., 2012) or dwell time (Yi et al., 2014), observed in platforms like social media or search engine result pages. These data are often paired with visual information under a supervised learning setting (Yang et al., 2022). While these behavioral signals act as proxies to cognitive responses to salient features, they may not always capture the nuanced cognitive preferences of individuals.

In this work, we propose an alternative approach, *cognition-supervised learning*, to obtain human responses toward visual information without any reliance on manual labels or behavioral data. Our approach relies on natural reactions of human cognition as a preference signal measured via electroencephalogram (EEG) as evoked in response to human visual perception. That is, a participant is only looking at visual information and the EEG signals of the participant's brain activity evoked in response to visual perception are recorded. The EEG data are then used within a self-supervision setting to learn representations that reflect the salient semantic visual features that cause differences in human cognitive responses.

Historically, integrating brain responses into machine learning has been challenging. Previous research on visual saliency detection using brain responses often relied on manually labeled data (Pinto et al., 2023; Zheng et al., 2020; Chen et al., 2021) or fine-tuning pre-trained models that are trained with manually labeled data (Santamaría-Vázquez et al., 2020; Cooney et al., 2019; Elsayed et al., 2021; Takagi & Nishimoto, 2022). On the other hand, unsupervised methods often underperform

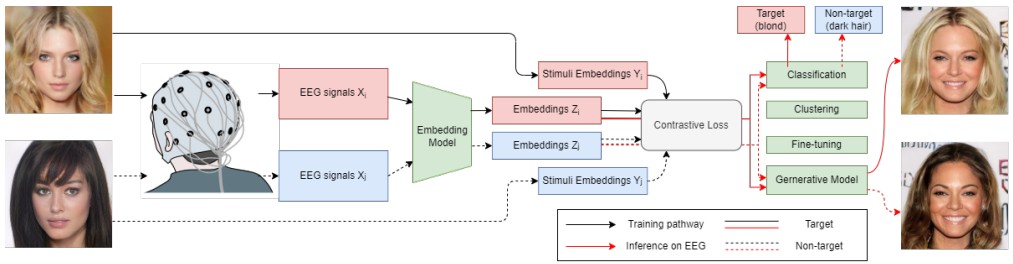

Figure 1: Illustrations of proposed cognition-supervised learning. EEG responses to visual stimuli are used to train embedding models, with CLIP loss using the vectors that were used to generate the stimuli images. The learned EEG embeddings can be applied in downstream tasks such as clustering, classification, fine-tuning personalized models and condition generative models. No manual annotation is needed during training or inference.

on brain data due to the inherent noise and complexity of brain data (Nishimoto et al., 2020). Even in supervised binary classification, when only single-trial data is available and the visual stimuli is complex, the typical accuracy often hovers around 0.7, for example, 0.78 for classifying human faces against objects (Lawhern et al., 2018), around 0.7 for within-subject and around 0.4 for cross-subject (Lawhern et al., 2018), 0.708 for subsets of ImageNet (Ahmed et al., 2021b) and below 0.6 for text stimuli (Eugster et al., 2014). Furthermore, a line of earlier research tackling similar tasks has also been questioned for confounded datasets (Li et al., 2020). Consequently, the problem of directly using brain signals as a source for the supervision of machine learning models has remained unsolved.

Our novel approach learns a representation of the visual saliency perceived by the brain directly using unlabeled EEG data, contrasted with visual stimuli, as the primary source of supervision. The model is designed to distinguish target and non-target saliency based on participants' brain responses.

Using the model, we address the following two primary research questions:

**RQ1:** Can representations of semantic saliency be learned directly using EEG data as a supervision signal?

**RQ2:** Do the learned representations of semantic saliency accurately reflect the salient features in downstream tasks?

Furthermore, to support and encourage further research in this domain, we are releasing an open, anonymized EEG dataset from 30 participants. This dataset, complete with well-defined semantic saliency detection tasks, aims to catalyze advancements in cognition-supervised models.

In summary, our primary contributions include:

- A novel approach for contrastively training models using cognitive EEG responses to visual stimuli, to learn representations of semantic visual saliencies.
- A new open and anonymized EEG dataset from 30 participants, accompanied by a comprehensive codebase to foster research in cognition-supervised models.

## 2 RELATED WORK

In recent years, the integration of brain signals and machine learning has received considerable attention due to its potential to enhance the performance and interpretability of machine learning algorithms. Among various brain-computer interface devices, electroencephalography (EEG) signals have emerged as a popular modality, providing rich yet noisy information for supervised machine learning models. EEG offers advantages such as a non-intrusive setting, high temporal resolution, and cost-effectiveness. However, EEG signals are inherently limited in spatial resolution and are prone to artifacts and noise caused by subject movements, which can significantly hinder the performance of EEG-based machine learning models, particularly those involving cognitive processes

such as visual semantic saliency recognition. The low spatial resolution of EEG signals may pose challenges in accurately capturing the precise localization of neural activity associated with visual cognition, while noises can introduce additional distortions to the relevant cognitive signals of interest. Decoding EEG signals have enabled a wide range of applications, including motor imagery (Altuwaijri et al., 2022; Padfield et al., 2019), emotion recognition (Al-Nafjan et al., 2017; Huang et al., 2019), mental workload assessment (Aricò et al., 2016; Riccio et al., 2011), and sleep stage classification (Chen et al., 2018; Chambon et al., 2018). The foundation of these EEG-based machine learning applications lies in the supervised EEG classification models, which enable the effective utilization of brain data in various contexts.

However, traditional approaches to supervised machine learning rely on manual annotations, which pose challenges in terms of cost and subjectivity. Manual annotations require domain experts to label large amounts of data, making the process time-consuming and resource-intensive. Furthermore, the subjectivity of human annotations introduces inter-annotator variability, compromising the reliability and consistency of annotations, particularly for subjective phenomena like emotions and mental states. To address these limitations, there is a growing need for unsupervised and self-supervised approaches that leverage EEG as supervisory signals to train machine learning models.

Recent research has explored the direct utilization of brain signals as supervisory signals for machine learning models (DelPreto et al., 2020; Chen et al., 2018; Chambon et al., 2018). The self-supervised learning with EEG data may provide potentially more objective and quantifiable measures of brain activity, which in turn leads to more reliable and cost-effective annotations compared to traditional methods that require expert knowledge for manual annotation. Moreover, the real-time capture of brain responses enhances the adaptability and robustness of machine learning models, enabling them to dynamically respond to changes in brain states.

A series of earlier studies on EEG-based image reconstruction suffer from confounded EEG data due to specific experimental block designs. This includes the EEG-GAN approach (Palazzo et al., 2017; Spampinato et al., 2017), Thoughtviz (Tirupattur et al., 2018), Brain2image citepkavasidis2017brain2image, EEG-ChannelNet (Palazzo et al., 2020), and numerous subsequent research on the same datasets such as EEG2IMAGE (Singh et al., 2023), DM-RE2I (Zeng et al., 2023), NeuroGAN (Mishra et al., 2023), and GDN-GAN (Khaleghi et al., 2022). However, subsequent analyses (Li et al., 2020; Ahmed et al., 2021b;a) have identified a critical flaw in these approaches: the block design in data collection introduces temporal correlations between the presentation order of stimulus class and the experiment's duration. Replication attempts have suggested that models were learning to recognize stimuli order rather than genuine cognitive reactions to stimuli.

In parallel, contrastive learning methods have gained significant attention in the broader field of machine learning (Wang & Qi, 2022; He et al., 2020; Xu et al., 2022; Chen et al., 2020a; He et al., 2022; Radford et al., 2021; Gunel et al., 2021). Contrastive learning aims to learn robust and useful representations without explicit annotations by maximizing the agreement between constructed positive pairs (similar samples) and minimizing the agreement between negative pairs (dissimilar samples). While the success of contrastive learning methods has been demonstrated in areas such as large language models (Radford et al., 2021), image embeddings (Jaiswal et al., 2020), and audio data (Saeed et al., 2021), its potential in EEG-based machine learning models remains largely unexplored. The similar methods has been applied to EEG modality as well, for instance sleep stage classification (Jiang et al., 2021), emotion recognition (Mohsenvand et al., 2020) and pathology screening (Banville et al., 2021). These self-supervised contrastive learning methods heavily depend on carefully designed artificial data augmentation transformations or random combinations of weak transformations. However, the efficient data augmentation transforms for contrastive learning with EEG data are still largely unknown. As discovered in a recent work (Jiang et al., 2021), a wrong choice of transformation can decrease the test accuracy from $82.90\%$ to $48.15\%$.

On the other hand, contrastive learning methods on a single modality discard potentially useful information from other modalities. This issue was addressed by supervised contrastive learning (Gunel et al., 2021), which groups augmented image pairs with the help of labels. Another recent work (Xu et al., 2022) proposed a hierarchical semantic alignment strategy to model the semantic similarity between images. Additionally, a multimodal contrastive training approach (Yuan et al., 2021) adopted multiple loss functions to exploit the intrinsic data structure within each modality. Our work build on top of the well-known language supervision approach CLIP (Radford et al., 2021) which learns representation from paired text and image data that aligns between two modalities, our embedding

model aligns representations from paired EEG and visual stimuli, which comes effortlessly from the data collection steps. Instead of decoding EEG signals to the categorical or simple stimuli, the contrastive methods bridges EEG to high-dimensional stimuli. Analogous to CLIP which classifies itself as natural language supervision, we consider our approach as cognition supervised learning.

One recent study (Schneider et al., 2023) explored non-linear methods for learning a consistent latent space of joint behavior and neural data across subjects and evaluated the approach using various animal data. However, it is important to note that all the data were collected with intrusive implanted electrodes or probes, which, unlike EEG signals, are more difficult to obtain for human subjects. The study claimed movie frame reconstruction, but it focused on recovering the order of movie frames and assumes the frames remain the same between the training and testing sets, which limits its generalizability.

Therefore, there exists a research gap for effective cognition-supervised learning. To bridge this gap, we propose a novel method that utilizes the contrast between EEG data and stimuli as a supervision signal. Our approach combines the merits of label-free learning with EEG data, while also incorporating the stimuli information to ensure effectiveness even when the available amount of EEG data is limited and insufficient for self-supervision based solely on EEG signals.

## 3 METHODS

### 3.1 DATA COLLECTION AND PREPARATION

In order to investigate the feasibility of cognition-supervised learning, we conducted neurophysiological experiments to collect EEG responses to generated visual stimuli. Our experiments are accepted by the ethical review board of *anonymous organization* and fully comply with declaration of Helsinki. Refer to our ethics statement for details.

**Visual Stimuli Preparation.** We choose generated face images as visual stimuli, as human is known to respond strongly to facial stimuli (Vuilleumier et al., 2001). We opted for generated images over real images to better control variances in semantics and confounding visual features, avoiding brain responses associated with recognition effect. The generated homogeneous dataset also allows strict semantic-level evaluation in generative tasks. A random sample of 70,000 images was generated from a progressive GAN[1] (Karras et al., 2018) pre-trained on the CelebA dataset (Liu et al., 2015). The images were manually screened to ensure realism and the absence of visual artifacts. These images were then grouped into eight categories based on their semantic saliency: smiling, not smiling, female, male, young, old, dark hair, and light hair (blond). Further details on stimuli preparation can be found in Sec. A.1.

**Participants.** Neurophysiological data were collected from thirty participants (self-reported 13 as female and 17 as male, mean age 28 years (SD = 7.14, Min = 18, Max = 45)) at *anonymous organization* . The participants were healthy with normal or corrected-to-normal vision.

**Apparatus, Tasks, and Procedure** Participants were sequentially presented with eight recognition tasks, each task corresponding to a semantic saliency group (e.g., female, smiling, etc.). During each task, all stimuli presented were assigned a binary label based on semantic saliency. For example, during the task "smile", participants were shown smiling faces (target) or non-smiling faces (non-target). Participants were instructed to only observe the presented images and make a mental note whenever they saw an image that matched the task description. No other mental or physical inputs were asked from the participants. Twenty target and twenty non-target images were shown in random order during each iteration of the task. Stimuli were presented following a rapid serial visual presentation (RSVP) procedure at the rate of 500 ms, such that stimuli were presented sequentially. Before each task, participants completed a demonstration task to ensure that they understood the experiment. Only in this demonstration task, they were asked to select the images that contained the semantic feature of interest.

**Data Preprocessing.** After the data acquisition, standard signal cleaning procedures were employed (Luck, 2014) to improve the signal-to-noise ratio. These were restricted to only automatic

---

[1] `https://github.com/tkarras/progressive_growing_of_gans` under attribution-noncommercial 4.0 international (cc by-nc 4.0) license

procedures that do not require any additional labeled data, including a band-pass filter 0.2-35 Hz, time-locking to *epochs* ranging from -200 to 900 ms relative to stimulus onset, and eyeblink artifacts removal using a threshold-based heuristic. An average of 1144 epochs per participant remained after pre-processing and balancing target and non-target epochs. The data averaged over participants shows a typical P300 effect with amplified potentials for target stimuli starting at 250 ms after stimuli onset and lasting until 600 ms as illustrated in Figure 2. This finding confirms that, on population average, the experiment resulted in an expected ERP effect.

## 3.2 COGNITION-SUPERVISED LEARNING

Cognition-supervised learning leverages a fundamental observation that the human brain responds to differences in perception. This observation suggests that the contrast between visual stimuli and human brain responses can serve as a supervision signal to learn directly from the preference reflected in the cognitive processes. With this contrastive learning setting, it becomes possible to design a loss function that utilizes only EEG data and the stimuli, eliminating the need for manual annotations.

In order to achieve cognition-supervised learning, we propose a model that learns an embedding of EEG signals. For each stimulus image generated from a latent vector $Y \in \mathbb{R}^L$ of dimension $L$, we vectorize the epoched EEG response signal as $X \in \mathbb{R}^{C \times T}$, where $C$ is the number of channels and $T$ is the number of time steps in the sliced window.

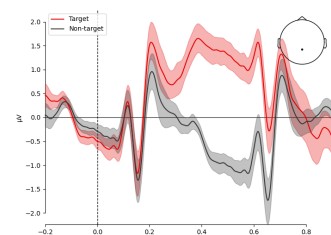

A straightforward approach to learning the embedding is to use a regression model $f_{reg} : \mathbb{R}^{C \times T} \to \mathbb{R}^L$ to reconstruct the stimulus vector from the EEG inputs. However, we found that this method overfits to noise and has poor generalizability. Additionally, it is not practical to reconstruct the entire stimulus vector from the EEG since only the salient semantic features and major face attributes are recognized by the participant. Therefore, we use a noise contrastive CLIP loss (Radford et al., 2021) and aim to learn an embedding to represent the semantic saliency perceived by the participant.

Figure 2: The average event-related potentials (ERPs) over the participant population at the Pz electrode for target and non-target stimuli. The ERPs reflect a P300 effect.

We propose a method for training an embedding model $f_{\text{embed}} : \mathbb{R}^{C \times T} \to \mathbb{R}^L$. Given an EEG signal $X$ and its corresponding stimuli vector $Y$, we construct a set of stimuli vectors $Y_i$ as negative stimuli vectors, where $i \in \{2, 3, \ldots, N\}$. The negative set is sampled from the remaining stimuli vectors in the dataset while avoiding duplication of $Y$ and $Y_i$. We add $Y_1 := Y$ as the positive sample.

The model $f_{\text{embed}}$ is trained to predict the probability $\hat{p}_j = \mathbb{P}[Y_j = Y]$ by computing the dot product between $Z := f_{\text{embed}}(X)$ and each $Y_j$, followed by a Softmax function. The probability function is given by

$$\hat{p}_j = \frac{e^{\langle Z, Y_j \rangle}}{\sum_{j'=1}^{N} e^{\langle Z, Y'_j \rangle}} \tag{1}$$

where $\langle \cdot, \cdot \rangle$ denotes the inner product.

We train $f_{\text{embed}}$ using cross-entropy between $p_j$ and $\hat{p}_j$ with $p_j = 1$ if and only if $j = 1$ otherwise $p_j = 0$. The loss function can be simplified as

$$L_{\text{CLIP}}(p, \hat{p}) = -\langle Z, Y \rangle + \log(\sum_{j=2}^{N} e^{\langle Z, Y_j \rangle}) \tag{2}$$

## 3.3 STRUCTURE OF THE MODEL

In order to account for the inter-subject variability, while learning the intrinsic structures of EEG signals, we utilize a deep neural network $f_{\text{embed}}$, that takes as input the raw vectorized EEG sig-

nals. In addition, a one-hot encoded vector representing the corresponding participant is given. The network outputs an embedding vector $Z$ that is of the same length as the stimulus vector $Y$. The structure of the network consists of two parts: (1) a participant-specific convolution matrix and (2) a sequence of fully connected layers. See Sec. A.2 for detailed specifications.

**Participant-Specific Matrix.** To create a unified model that can incorporate the variability between participants, we adopt the approach introduced in (Défossez et al., 2022). A participant-specific layer is included at the beginning of the network. This layer consists of a trainable matrix of size $C \times C$ for each participant. This $C \times C$ matrix is multiplied by the vectorized $C \times T$ EEG signal by channels. We initialize the matrix randomly, close to the identity matrix with small random noise added.

**Data Augmentation.** To improve the generalization ability of our model and avoid overfitting, we use random data augmentation during training. First, we first multiply each EEG vector $x \in \mathbb{R}^{C \times T}$ with a random vector $c \in [0.95, 1.05]^C$. Next, we randomly crop and resize the vector into the original shape. For the validation set, we use the central interval without random multiplication.

## 4 EXPERIMENTS

We evaluate the efficiency of cognitive supervision using four different evaluation strategies. First, we conduct unsupervised clustering of the embedding space to reveal whether it corresponds to the target and non-target saliency. Second, we follow a common linear evaluation protocol (He et al., 2016; Oord et al., 2018; Kolesnikov et al., 2019; Bachman et al., 2019; Cole et al., 2022; Da Costa et al., 2022) to study whether linear classifiers trained on top of the saliency embeddings perform better than those trained on data that is not supervised from human cognition. Third, we examined personalized tuning of the embedding models. Fourth, we conduct a qualitative evaluation by visualizing the outputs of the cognition-supervised predictions via generative adversarial networks. Detailed experimental setups and hyperparameters are described in Sec. A.3.

### 4.1 UNSUPERVISED CLUSTERING

**Evaluation procedure.** The dataset in each task should have two clusters, the target, and the non-target cluster because the stimuli are deliberately ensured to either contain the task-specific semantic saliency or not. To obtain the two distinct clusters, we trained the embedding model on the entire dataset and ran KMeans with $k = 2$ on the frozen embeddings, then cluster with a higher averaged P300 effect was selected as the Target cluster $C_T$, while the other cluster as the Non-target cluster $C_N$. For evaluation only we used the explicit stimuli labels to compute the clustering accuracy.

**Control models.** We consider three control models that apply KMeans clustering to different inputs: (1) stimuli vectors; (2) flattened EEG signals; (3) concatenated EEG signals and paired stimuli vectors. We followed the standard protocol of enumerating all possible cluster permutations and reporting the highest accuracy achieved for control models.

**Results** The clustering accuracies, presented in Table 1, indicate that our embedding model consistently outperformed the control models across all tasks with substantial improvements. Furthermore, the results verify that the learned embedding captures the salient features perceived by the participant.

### 4.2 LINEAR EVALUATION

**Evaluation procedure.** To evaluate the efficiency of the learned saliency representations, we follow the commonly used linear evaluation protocol, by training a linear classifier on top of the frozen embeddings. The dataset is randomly split into a training set and a testing set with disjoint sets of stimuli. We then train our contrastive embedding model on the training set and then compute the embeddings with frozen model weights. A single-layer binary classifier $C(\cdot) : \mathbb{R}^{512} \to \{0, 1\}$ is trained on the embeddings from the training set using the explicit labels of stimuli images. The classifier is then evaluated on the test set using the labels with classification accuracy.

**Control models.** To provide a basis for comparison, we also consider three control models as the baseline. The first is a well-known supervised EEGNet (Lawhern et al., 2018) structure to estimate

Table 1: Clustering accuracies on all tasks with different inputs for KMeans.

| KMeans input | female | male | blond | darkhaired | smiles | nosmile | old | young | Mean |
|---|---|---|---|---|---|---|---|---|---|
| stimuli vectors | 0.591 | 0.511 | 0.548 | 0.501 | 0.501 | 0.506 | 0.543 | 0.545 | $0.531 \pm 0.030$ |
| EEG signals | 0.568 | 0.512 | 0.559 | 0.521 | 0.565 | 0.503 | 0.520 | 0.549 | $0.537 \pm 0.024$ |
| concatenated | 0.545 | 0.549 | 0.580 | 0.605 | 0.503 | 0.535 | 0.541 | 0.501 | $0.545 \pm 0.033$ |
| saliency embedding | **0.816** | **0.786** | **0.803** | **0.773** | **0.745** | **0.720** | **0.626** | **0.702** | **0.746** $\pm 0.059$ |

Table 2: Linear classification accuracies for all models. The train/test split is the same for all models.

| Tasks | female | male | blond | darkhaired | smiles | nosmile | old | young | Mean |
|---|---|---|---|---|---|---|---|---|---|
| EEGNet | **0.773** | 0.687 | 0.727 | 0.673 | **0.728** | 0.678 | **0.655** | **0.670** | $0.699 \pm 0.037$ |
| LDA | 0.596 | 0.539 | 0.585 | 0.573 | 0.563 | 0.549 | 0.522 | 0.538 | $0.558 \pm 0.024$ |
| random control | 0.485 | 0.488 | 0.503 | 0.543 | 0.471 | 0.501 | 0.471 | 0.502 | $0.496 \pm 0.022$ |
| contrastive embedding | 0.765 | **0.725** | **0.735** | **0.731** | 0.724 | **0.685** | 0.614 | 0.658 | **0.704** $\pm 0.046$ |

the upper limit of performance for the cognition-supervised models and highlights the difficulties of the task. The second is a linear discriminant analysis model (LDA) (Blankertz et al., 2011) to estimate the separability of raw EEG signals. Both control models are trained on the raw EEG signals and the explicit labels. The third baseline model is a randomly permuted cognition-supervised EEG classifier to determine a lower bound performance, in which the pairs of EEG signals and stimuli vectors are shuffled so that the pairs are broken.

**Results.** Table 2 shows the mean accuracies of all models for each task. The linear classifiers on saliency embeddings consistently outperform the random baseline and the LDA models, indicating that the learned embeddings were effective in disentangling semantic features. Furthermore, we observed that the mean accuracy across all tasks is higher than that of the EEGNet, which suggests that the learned embeddings successfully reduced the high dimensionality of raw EEG signals while preserving the saliency perceived by the participant. It is worth noting that, the embedding model is trained without labels and the supervised linear classifier on top of it is expected to have relatively lower performance compared to a completely supervised model, as shown in (Chen et al., 2020b).

## 4.3 PERSONALIZED MODEL EVALUATION

**Evaluation procedure.** In order to extend the utility of our model to reflect the cognitive responses of an individual, we evaluate fine-tuned personalized models. For each of the 30 participants, we first train a base model with the EEG data from the other 29 participants. Next, the data from the target participant is split into a 5-fold training set and a test set. We then freeze the base model weights except for the participant-specific matrix of the target participant. This matrix is randomly initialized then fine-tuned on this single-participant training set and the frozen saliency embeddings from other participants are clustered to assist in selecting the target cluster.

**Control models.** For comparison, we also evaluated two control models. The first control model is the base model evaluated on the test set without fine-tuning. The target participant matrix is set to the identity matrix. The second control model was fine-tuned on randomly shuffled training data, breaking the pairs of EEG signals and stimuli vectors.

**Results.** Table 3 shows the mean clustering accuracy on the test set and reports the mean of 5-fold validation across all participants. The consistently improved accuracy of personalized models over the base models demonstrates that our method is capable of adapting to different individuals and generalize. Moreover, the base model, which is not trained on the personal data, achieved high clustering accuracy compared to the random control model. This promising ability of zero-shot prediction of our embedding model suggests its potential to learn robust representations and its flexibility to learn subjective information from the cognitive signals of individuals.

Table 3: Mean clustering accuracies of all personalized models.

| Tasks | female | male | blond | darkhaired | smiles | nosmile | old | young | Mean |
|---|---|---|---|---|---|---|---|---|---|
| random control | 0.538 | 0.509 | 0.532 | 0.529 | 0.484 | 0.493 | 0.520 | 0.520 | $0.516 \pm 0.018$ |
| base model | 0.740 | 0.662 | 0.697 | 0.677 | 0.697 | 0.654 | 0.613 | 0.652 | $0.674 \pm 0.036$ |
| personalized model | **0.743** | **0.667** | **0.702** | **0.683** | **0.701** | **0.659** | **0.619** | **0.657** | **0.679** $\pm 0.035$ |

## 4.4 QUALITATIVE EVALUATION VIA GENERATIVE VISUALIZATION

**Generative Visualization of Salient Features.** In order to provide an intuitive understanding of the inter-subject variations, we visualize the embeddings for a qualitative evaluation. We sample a set $S$ of candidate stimuli vectors which can be the set of stimuli vectors in the training set, or a fresh set of randomly sampled vectors from the noise distribution used in the generative model. Each embedding $Z$ in a cluster $C$ is mapped to one of these stimuli $v_Z = \mathrm{argmax}_{Y \in S} \langle Z, Y \rangle$, and use the mean $M_C = \frac{1}{|C|} \sum_{Z \in C} v_Z$ to represent the cluster. We visualize it using the pre-trained generative model. For each task, we expect the image generated from $M_{C_\mathrm{Target}}$ to contain salient task-specific semantic features and $M_{C_\mathrm{Non-target}}$ to have the opposite semantic saliency.

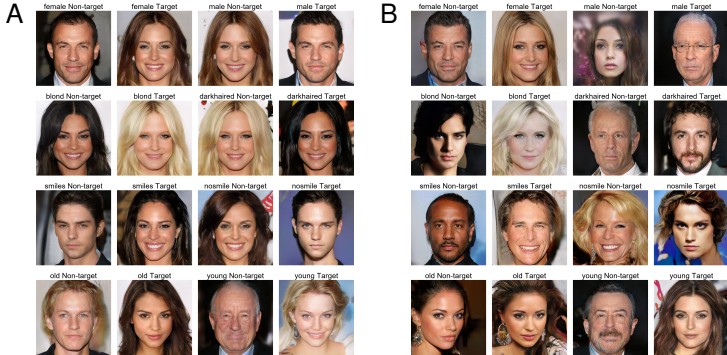

Figure 3: Visualization of clusters after mapping to stimuli vectors from **A** the training set and **B** randomly sampled stimuli vectors. For each task and each cluster, one image is generated by the mean mapped vector.

**Results.** The generated images with the stimuli set and the randomly sampled set are shown in Figure 3. Between the images from $M_{C_\mathrm{Target}}$ and $M_{C_\mathrm{Non-target}}$, it can be clearly seen that the semantic difference in images is correlated to the semantic task given to the participants. In Figure 3A the candidate image vectors are the stimuli vectors from the training set. In Figure 3B, 600 randomly sampled 512 vectors that are not present in the training set, are used as candidate sets and the semantic difference matching the task is still present. The salient features are clearly present across the different tasks and not present in the opposite tasks (Figure 3A). The representations result in generated images in which the intended salient features are present even for randomly sampled candidates (Figure 3B). This indicates that the learned embeddings reflect the underlying signal from human cognition for generating task-specific salient features.

In addition, a visualization of the learned embedding space is in Sec. A.4 For each individual, we also visualized the subset of embeddings from a single participant similarly in Sec. A.5.

## 4.5 ABLATION ANALYSIS

An ablation study was conducted to study the effects of participant-specific matrices and data augmentation. In contrast to the full model, three variants of the models were trained: (a) $M_\mathrm{no\ matrix}$ that removes the participant-specific matrix; (b) $M_\mathrm{no\ augmentation}$ that removes data augmentation; (c) $M_\mathrm{base}$ that removes both.

Table 4: Ablation study on model variants.

| Tasks | female | male | blond | darkhaired | smiles | nosmile | old | young | Mean |
|---|---|---|---|---|---|---|---|---|---|
| full model | **0.816** | **0.786** | **0.803** | **0.773** | **0.745** | **0.720** | 0.626 | **0.702** | **0.746** $\pm$ 0.059 |
| $M_{\text{base}}$ | 0.768 | 0.632 | 0.682 | 0.688 | 0.657 | 0.595 | 0.617 | 0.655 | 0.662 $\pm$ 0.050 |
| $M_{\text{no augmentation}}$ | 0.748 | 0.673 | 0.690 | 0.688 | 0.653 | 0.632 | 0.623 | 0.663 | 0.671 $\pm$ 0.037 |
| $M_{\text{no matrix}}$ | 0.738 | 0.668 | 0.695 | 0.670 | 0.712 | 0.642 | **0.632** | 0.648 | 0.676 $\pm$ 0.034 |

The base model $M_{\text{base}}$ and $M_{\text{no matrix}}$ assumes that all data are collected from the same participant, and we select the cluster with the higher ERP effect as the target cluster. To minimize the differences caused by random cropping in data augmentation or other dimension changes, the base model $M_{\text{base}}$ and $M_{\text{no augmentation}}$ crops the EEG signals with fixed intervals as used in the test set.

Table 4 shows the accuracies of the full model and other variants for each task. The full model with the participant-specific matrix consistently yields improved accuracy over all model variants in all tasks except the task old. The two variant models $M_{\text{no matrix}}$ and $M_{\text{no augmentation}}$ both have improved mean accuracy over the base model. These results indicate the effectiveness of the participant-specific layer and data augmentation.

## 5 CONCLUSIONS

We set out to study whether machine learning models could be directly supervised by monitoring human cognition via EEG and utilize those data for cognition supervising models to learn representations of visual saliency. To this end, we asked two research questions that we reflect on below.

**Can representations of semantic saliency be learned directly using EEG data as a supervision signal?** We introduced a novel approach for contrastively training models supervised only by brain signals and show that it is possible to learn semantic visual saliencies from relevant signals contained in EEG. Our models correctly capture the semantic saliency without any explicit manual annotations in the process.

**Do the learned representations of semantic saliency accurately reflect the salient features in downstream tasks?** The performance of the learned models was evaluated in classification, clustering, and image generation tasks using facial image data. Our results showed improved performance across tasks competitive to classification models that are pre-trained and fine-tuned using large labeled datasets. The image generation results show that the models yield performance that has face validity in several tasks and can be competitive even against models utilizing manually provided supervision data.

**Limitations.** Today, most brain-computer interfacing research is still limited by low accuracy and convenience compared to conventional user interfaces. However, while our device setup is still restricted to laboratory experimentation and may not be readily usable by the public today, our results demonstrate that it is possible to develop human-in-the-loop learning systems that tap directly into human cognitive processing without a requirement for manual labeling or reliance on inaccurate and indirect manual annotations or implicit behavioral information. A clear limitation of our approach is that it only learns target or non-target saliency for an individual. That is, the individual's underlying task is simply detecting whether something salient appears in the visual information the participant is perceiving or not, rather than assigning a label for the salient feature. Therefore, our approach can not replace annotation scenarios that require inputting a label, but rather a recognition scenario where individual preferences are modeled, or the individual's task can be determined by other means. Examples of applicable scenarios could be the detection of images in image search results that match the query, the detection of CAPTCHA images, or image annotation where a saliency detection task is pre-determined for the participant. Despite restrictions on what EEG data can entail, many of our results indicate performance that bypasses the performance of models trained with conventional manual labels.

More generally, the presented approach opens avenues for human-in-the-loop systems that naturally integrate with human cognition and allow human-machine collaboration solely based on passive observation of brain signals.

**Reproducibility Statement** To improve the reproducibility of our experiments, a detailed specification is described in Sec. A.2. Additional details on neurophysiological data acquisition are included in Sec. A.1. The source code to train and evaluate and anonymized datasets will be released publicly to further ensure reproducibility.

**Ethics Statement** The research that has been documented adheres to the ethical guidelines outlined by the ICLR. The neurophysiological data acquisition and the follow-up experiments were approved by the ethical review board of social and behavioral sciences at *anonymous organization* and the protocols and consents comply with the declaration of Helsinki[2]. Informed consent was signed by each participant to acknowledge their rights. The participants were compensated with vouchers for the local cinema.

We demonstrated for the first time that machine learning could be self-supervised directly from human brain signals captured via EEG: cognitive supervision. The approach opens avenues for novel machine learning systems where human-in-the-loop interactions are implemented by monitoring cognitive reactions as they happen in the human brain when individuals perceive digital information. While this is a new, powerful supervision paradigm that can enhance machine learning, the approach also implies ethical concerns. The most significant ethical risks do not emerge from the recording technology itself. However, if wearable sensors capable of monitoring human cognition become more pervasive, the signals could be used beyond the original consent. This has become possible as our results show that the models do not anymore need labels and specific calibration to tasks, but they can be self-supervised from brain responses. For example, large-scale collection of sensitive signals and self-supervised alignment to visually perceived data that their users are exposed to may enable inference of human and crowd opinions toward a vast amount of digital information. We already observe such effects in our proof-of-concept experiments. For example, most of the images generated for the task young are females and most of the images generated for the task old are males in Figure A.9. These concerns call for ethical guidelines to support the broader adoption of this technology.

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

# A    APPENDIX

## A.1    NEUROPHYSIOLOGICAL DATA ACQUISITION

**Visual Stimuli Preparation.**    We use a progressive GAN pre-trained on the CelebA dataset to generate a random sample of 70,000 images. Raw images have a resolution of 1024 by 1024 pixels. These images are screened by several researchers to exclude images with visual artifacts, such as distorted faces or other clear signs of an artificial image, to prevent brain responses from being influenced by artifact recognition rather than semantic saliency.

**RSVP and EEG setup.**    An elliptic grey frame was positioned over all images to mask the background. The EEG data were recorded using 32 Ag/AgCl electrodes, arranged according to the 10–20 system, and connected to a QuickAmp USB (BrainProducts GmbH, Gilching, Germany) amplifier running at 2,000 Hz. Eye movements were detected (for artifact removal) using two pairs of bipolar electrodes for artifact detection (1 cm to the lateral canthi of the left and right eye, and 2 cm above and below the right pupil). The electrode placement is shown in Figure A.4. Specifically, we used 32 equidistant electrodes situated at FP1, FP2, F7, F3, Fz, F4, F8, FC5, FC1, FC2, FC6, T7, C3, Cz, C4, T8, TP9, CP5, CP1, CP2, CP6, TP10, P7, P3, Pz, P4, P8, PO9, O1, O2, PO10, Iz, within the 10% system.

**Data Preprocessing.**    After the data acquisition, standard signal cleaning procedures were employed to improve the signal-to-noise ratio. These were restricted to only automatic procedures that do not require any additional labeled data, including a band-pass filter in the frequency range 0.2–35 Hz and time-locking to (*epochs*) ranging from -200 to 900 ms relative to stimulus onset with baseline correction based on a pre-stimulus period of -200 to 0 ms. Eyeblink artifacts were removed using a threshold-based heuristic, where the threshold is set to the 200th largest mean absolute value of epochs over all channels, and clipped to range $[10, 80]$. An average of 1144 epochs per participant remained after pre-processing and balancing target and non-target epochs.

## A.2    SPECIFICATION OF MODEL ARCHITECTURE

In order to account for the inter-subject variability, while learning the intrinsic structures of EEG signals, we utilize a deep neural network $f_{embed}$, that takes as input the raw vectorized EEG signals. In addition, a one-hot encoded vector representing the corresponding participant is given. The

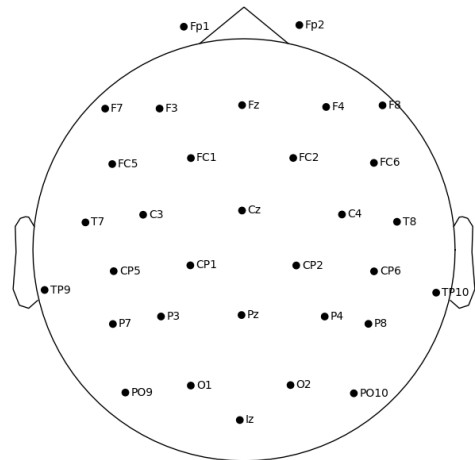

Figure A.4: Visualization of electrode placement.

network outputs an embedding vector $Z$ that is of the same length as the stimulus vector $Y$. The structure of the network consists of two parts: (1) a participant-specific convolution matrix and (2) a sequence of fully connected layers.

**Participant-Specific Matrix.** To create a unified model that can incorporate the variability between participants, we adopt the approach introduced in (Défossez et al., 2022). A participant-specific layer is included at the beginning of the network. This layer consists of a trainable matrix of size $C \times C$ for each participant. This $C \times C$ matrix is multiplied by the vectorized $C \times T$ EEG signal by channels. We initialize the matrix randomly, close to the identity matrix with small random noise added.

**Fully Connected Layers.** The embedding model consists of four fully connected layers, each of the first three layers has 2048 hidden nodes with an activation function LeakyReLU with an $\alpha = 0.3$. Each fully-connected layer is followed by a Dropout layer with a dropout rate of $0.5$. The last layer has 512 outputs with no activation.

**Data Augmentation.** For random data augmentations during training, we first multiply each EEG vector $x \in \mathbb{R}^{C \times T}$ with a random vector $c \in [0.95, 1.05]^{C}$. Next, we randomly crop and resize the vector into shape $x' \in \mathbb{R}^{C \times T}$ by selecting an interval $[l, r]$ with $l \in [0, \frac{T}{10}]$ and $r \in [T - \frac{T}{10}, T]$. For the validation set, we crop and resize the data with the fixed interval $[\frac{T}{20}, T - \frac{T}{20}]$, without random multiplication.

## A.3 EXPERIMENTAL SETUP

**Dataset.** Our dataset consists of EEG signal and stimuli vector pairs from 30 participants, comprising a total of 35490 pairs. To ensure that the individual factor is accounted for, we mixed all participant data while retaining a unique participant identifier to apply the participant-specific matrix. For the unsupervised task, we trained and evaluated our embedding model on the entire dataset. For linear classification tasks, we employed 10-fold validation by randomly splitting the dataset into training and testing sets and reported the mean of evaluation metrics.

**Hyperparameters and Hardware.** In all experiments, the brain embedding model is trained with Adam optimizers with an initial learning rate $1e - 4$, $\beta_1 = 0.9$, $\beta_2 = 0.999$, and a weight decay $1e - 4$. The mini-batch size is set to 256 in all experiments. We conducted all experiments on Tensorflow with Nvidia GeForce RTX 3070 Ti GPUs.

Each embedding model in the unsupervised clustering experiments and linear evaluation experiments and the base model in the personalized experiments are trained with 500 epochs. The personalized model is fine-tuned by 100 iterations by using the base model, with all other parameters frozen except the participant-specific matrix.

**Clustering Accuracy** For our embedding models, the clusters are compared for P300 effect by computing mean values of EEG signals in the window 200ms to 400ms, the cluster with a higher mean value was selected as the Target cluster $C_T$, while the other cluster was designated as the Non-target cluster $C_N$. For evaluation of the clustering accuracy only, we used the explicit labels $L_X = L_Y \in \{0, 1\}$ of each EEG signal-stimuli vector pair $(X, Y)$, where $L_X = 1$ indicates that the image from $Y$ contains the task saliency, and $L_X = 0$ otherwise. The clustering accuracy was computed as follows:

$$\text{Accuracy}(C_T, C_N) = \frac{\sum\limits_{X \in C_T} L_X + \sum\limits_{X \in C_N} (1 - L_X)}{|C_T| + |C_N|} \tag{3}$$

For control models, we followed the standard protocol of enumerating all possible cluster permutations and reporting the highest accuracy achieved. That is, if the two resulting clusters from a control model are denoted as $C_1$ and $C_2$, then the clustering accuracy for the model is computed as $\max\{\text{Accuracy}(C_1, C_2), \text{Accuracy}(C_2, C_1)\}$.

## A.4 VISUALIZATION OF EMBEDDINGS

We visualize the features space of the (1) raw EEG signals, (2) learned embeddings, and (3) image stimuli vectors using (a) Principal component analysis (PCA) and (b) t-SNE on task blond, as shown in Figure A.5. We observe that the distribution of EEG and stimuli are all entangled, but the learned embeddings have the target group (blond) and non-target group (dark hair) more separable. In addition, the largest principal component of the learned embeddings aligns with the task semantic, so that in Figure A.5C the targets and non-targets are separated to the left and right.

## A.5 ADDITIONAL VISUALIZATION OF EMBEDDINGS

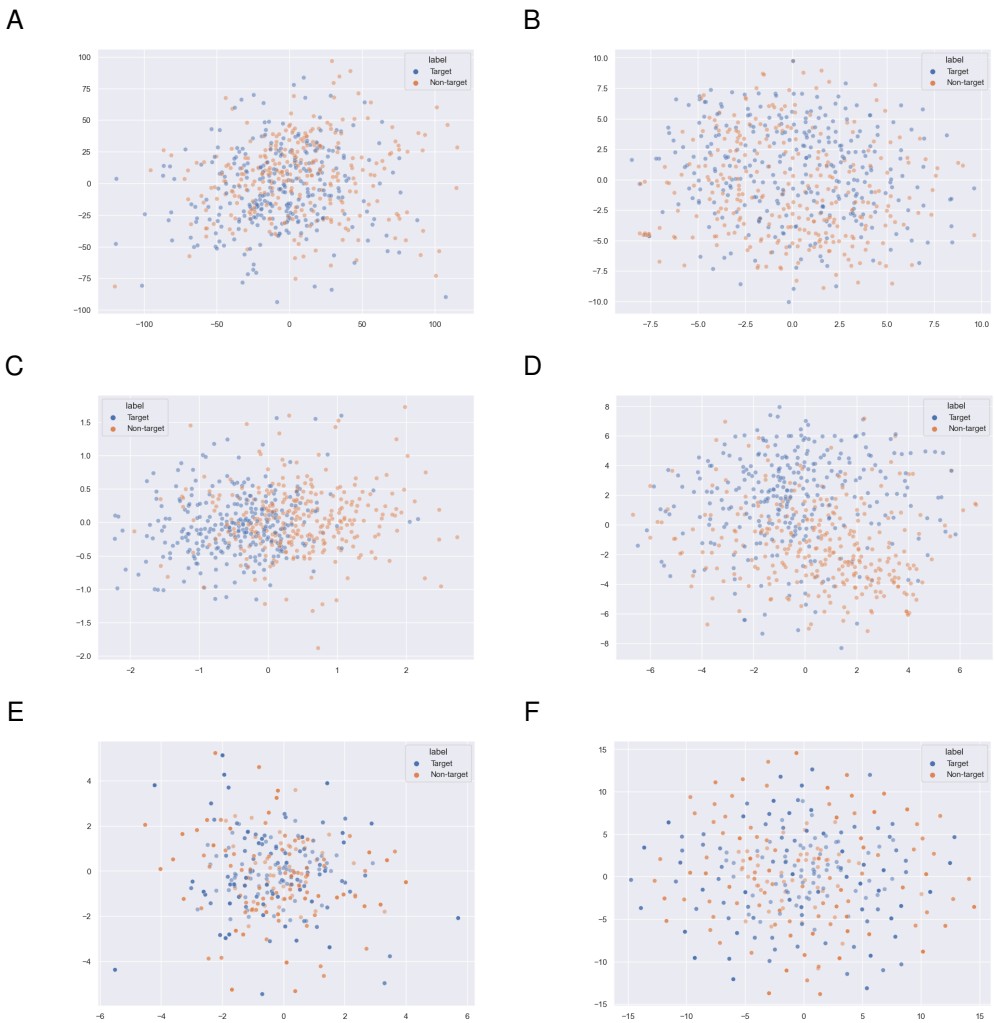

Figure A.5: Visualization of features space of **A** EEG by PCA, **B** EEG by t-SNE, **C** embeddings by PCA, **D** embeddings by t-SNE, **E** stimuli by PCA, **F** stimuli by t-SNE.

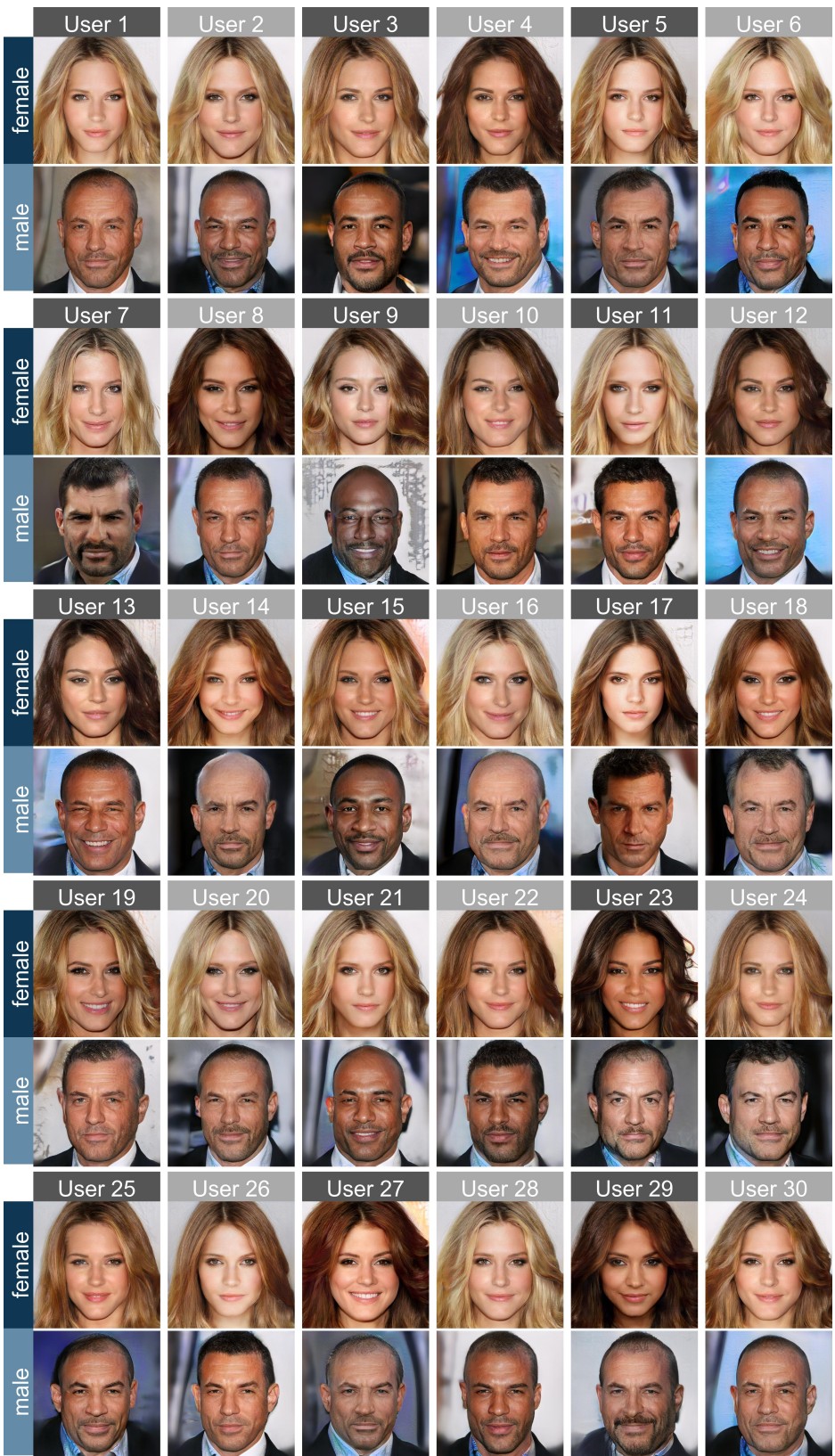

Figure A.6: Visualization of subsets of embeddings for each participant in task "female" and "male". The embeddings from the same participant are mapped to the stimuli image vectors.

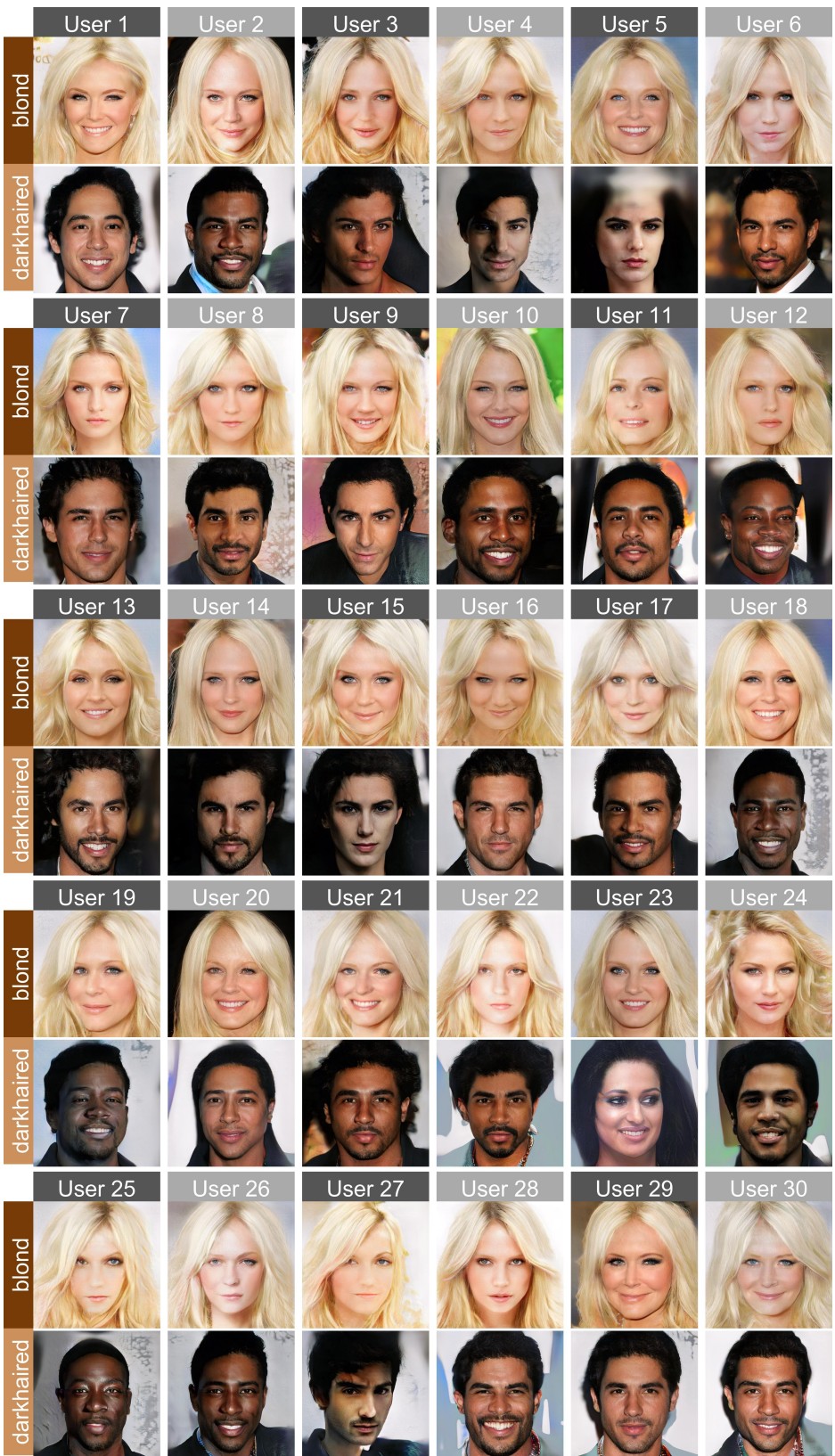

Figure A.7: Visualization of subsets of embeddings for each participant in task "blond" and "dark hair". The embeddings from the same participant are mapped to the stimuli image vectors.

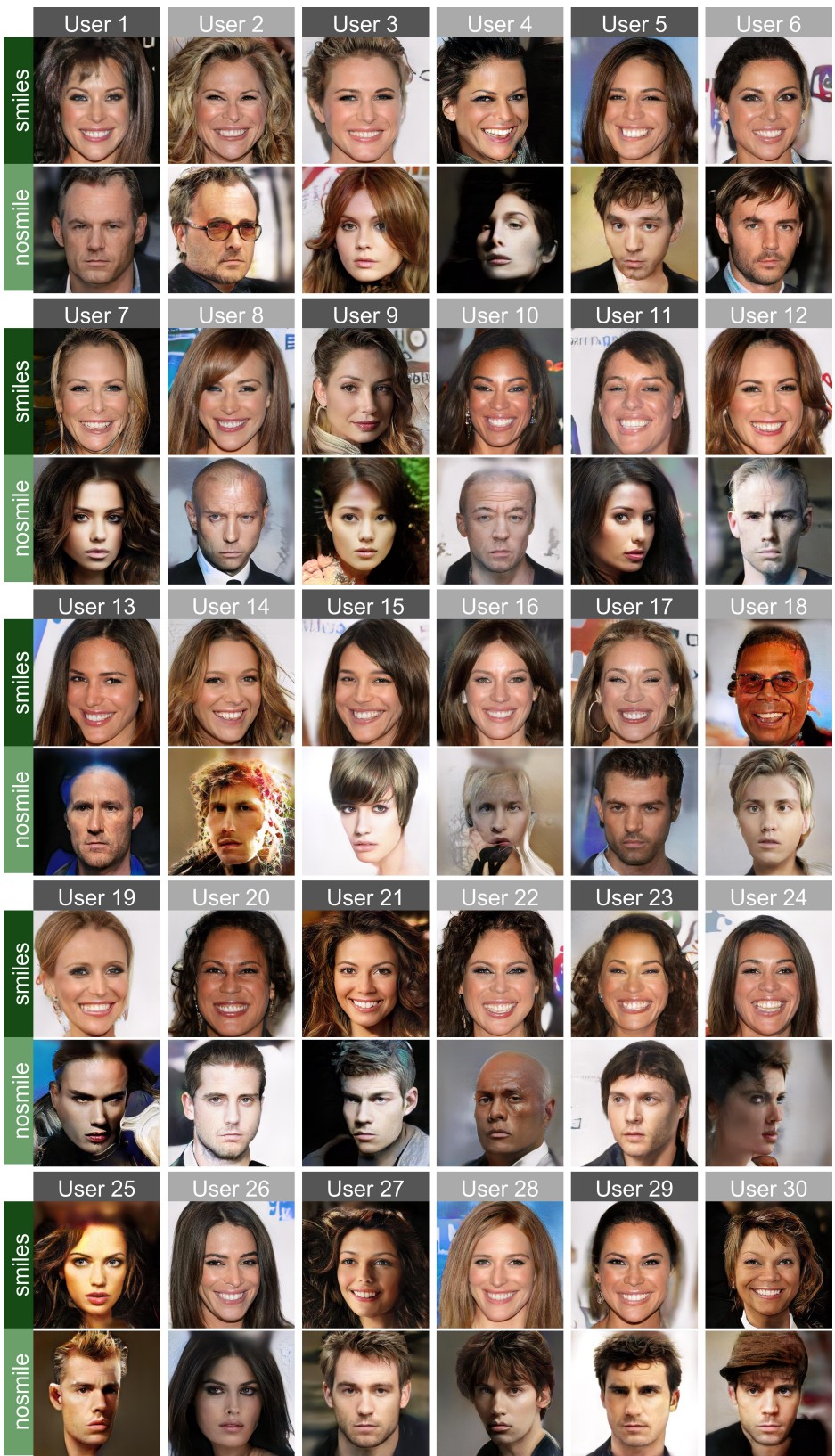

Figure A.8: Visualization of subsets of embeddings for each participant in task "smile" and "no smile". The embeddings from the same participant are mapped to the stimuli image vectors.

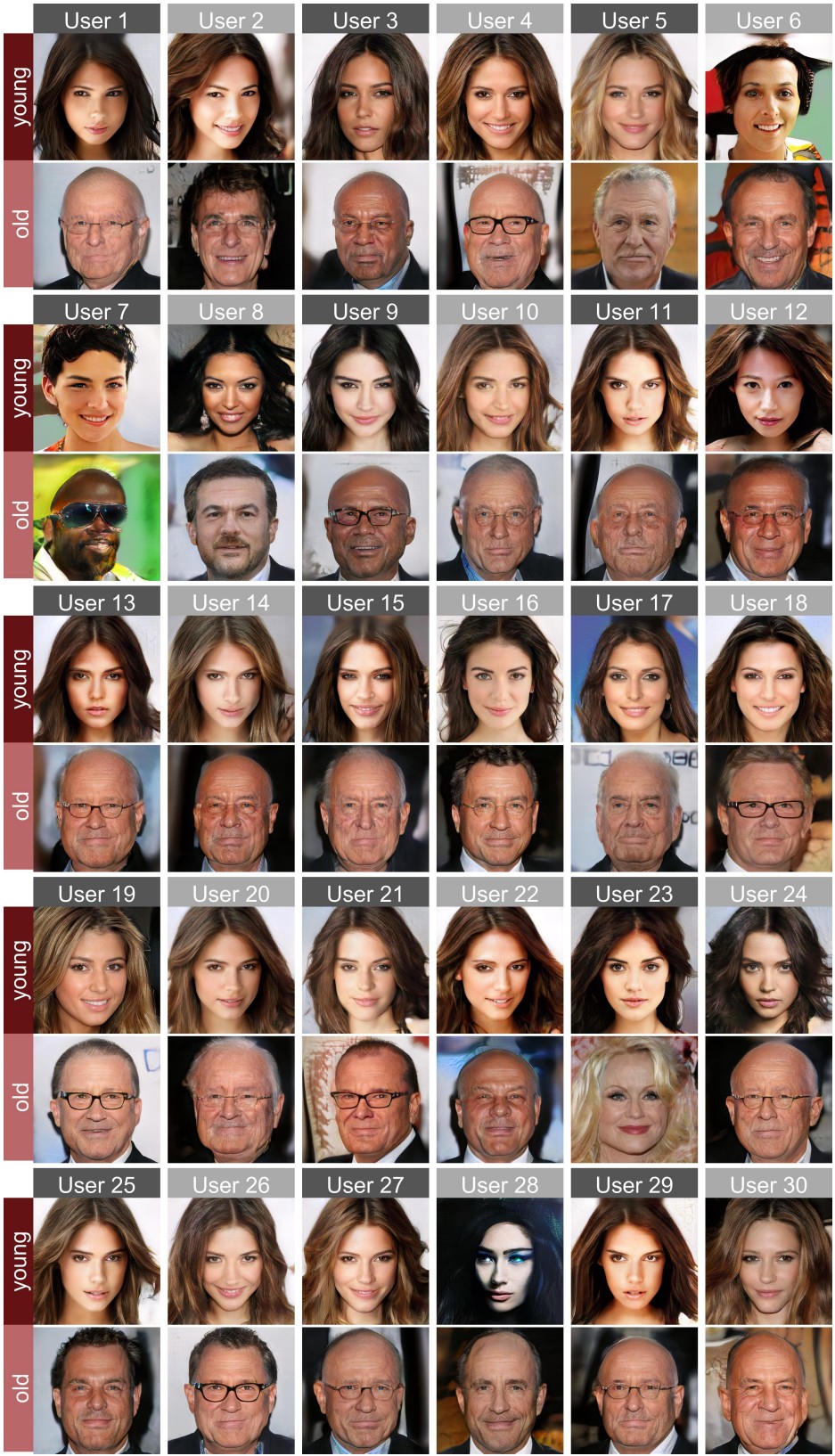

Figure A.9: Visualization of subsets of embeddings for each participant in task "young" and "old". The embeddings from the same participant are mapped to the stimuli image vectors.

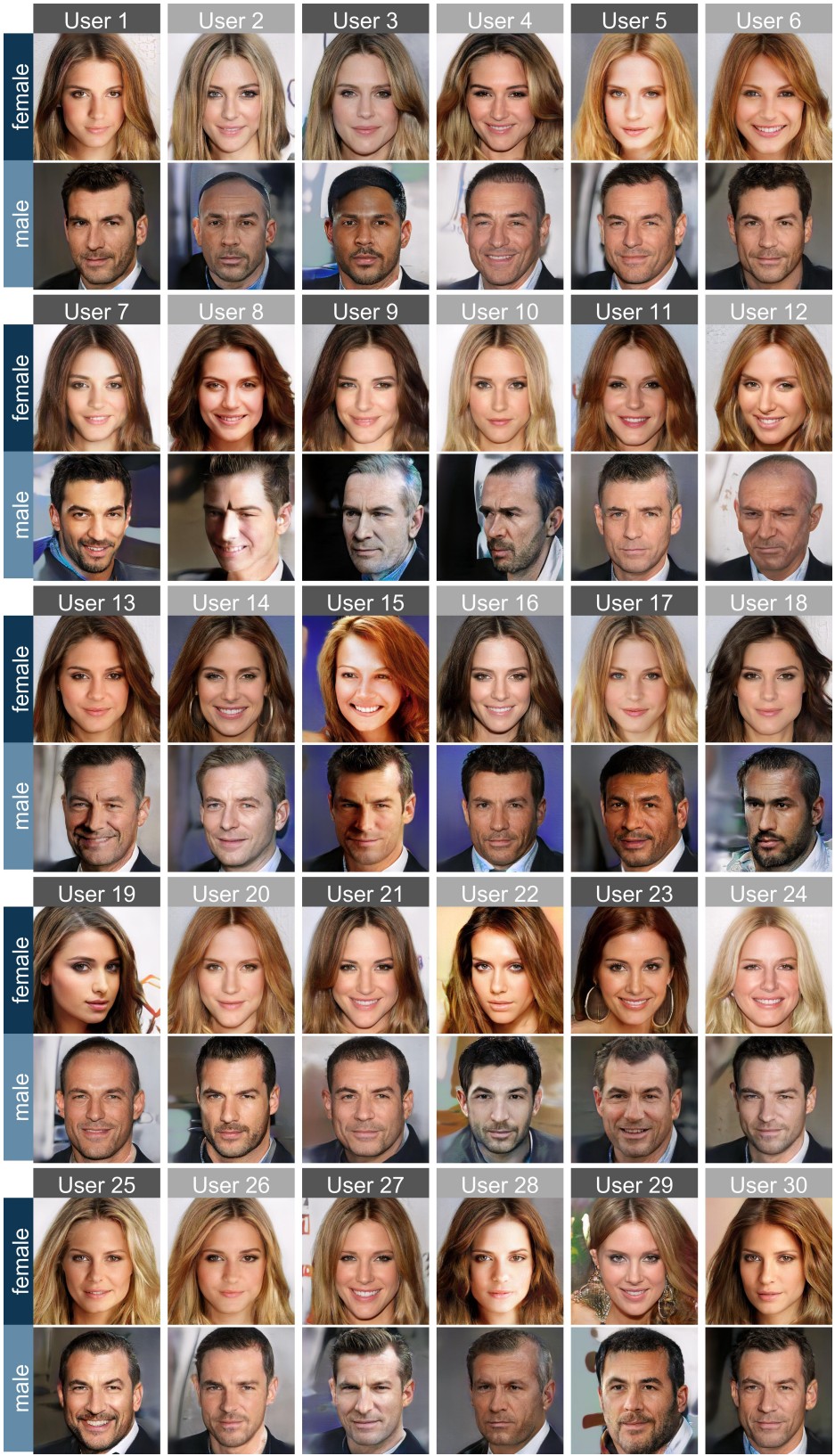

Figure A.10: Visualization of subsets of embeddings for each participant in task "female" and "male", by mapping embeddings to the randomly sampled candidate set.

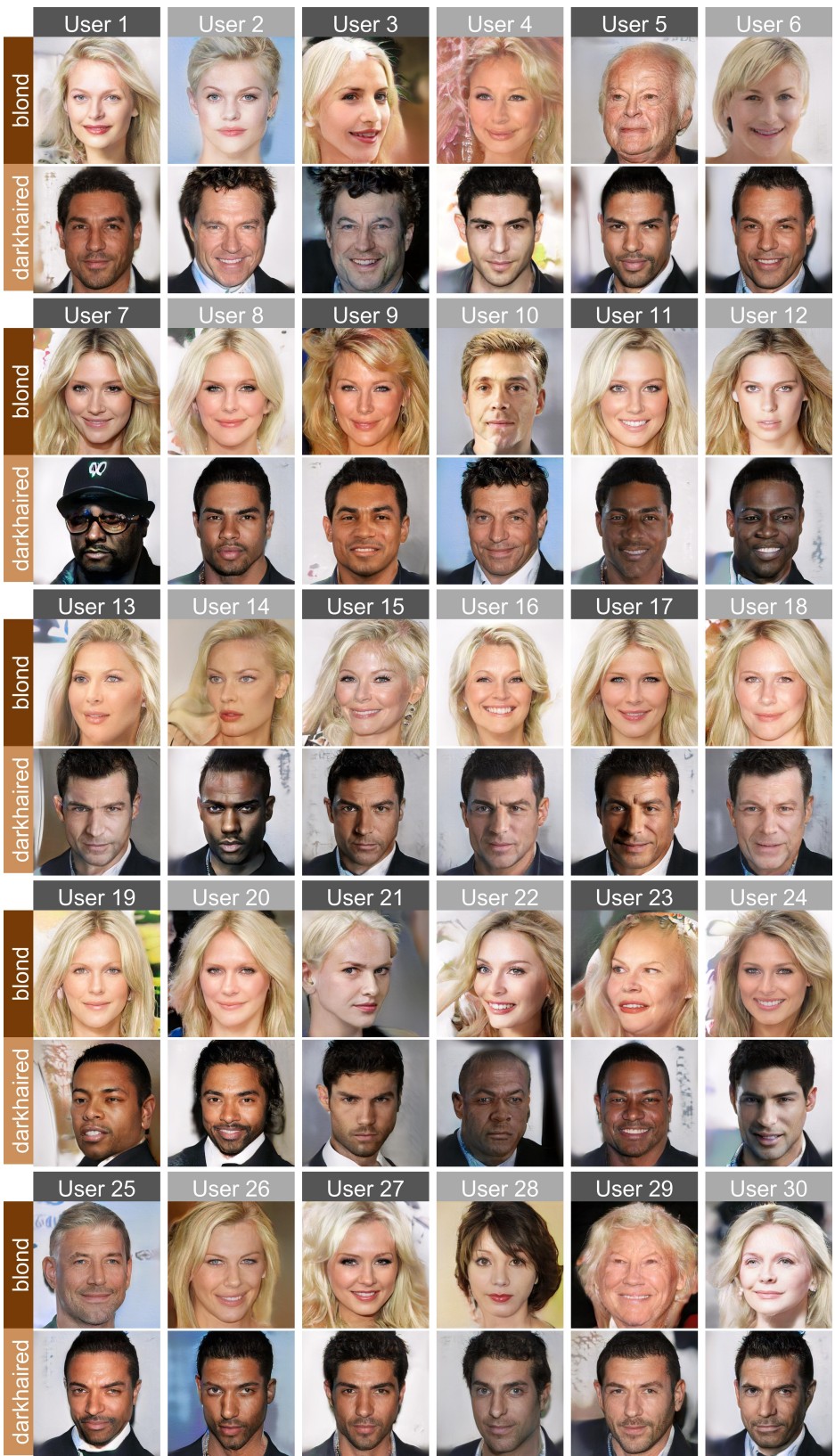

Figure A.11: Visualization of subsets of embeddings for each participant in task "blond" and "dark hair", by mapping embeddings to the randomly sampled candidate set.

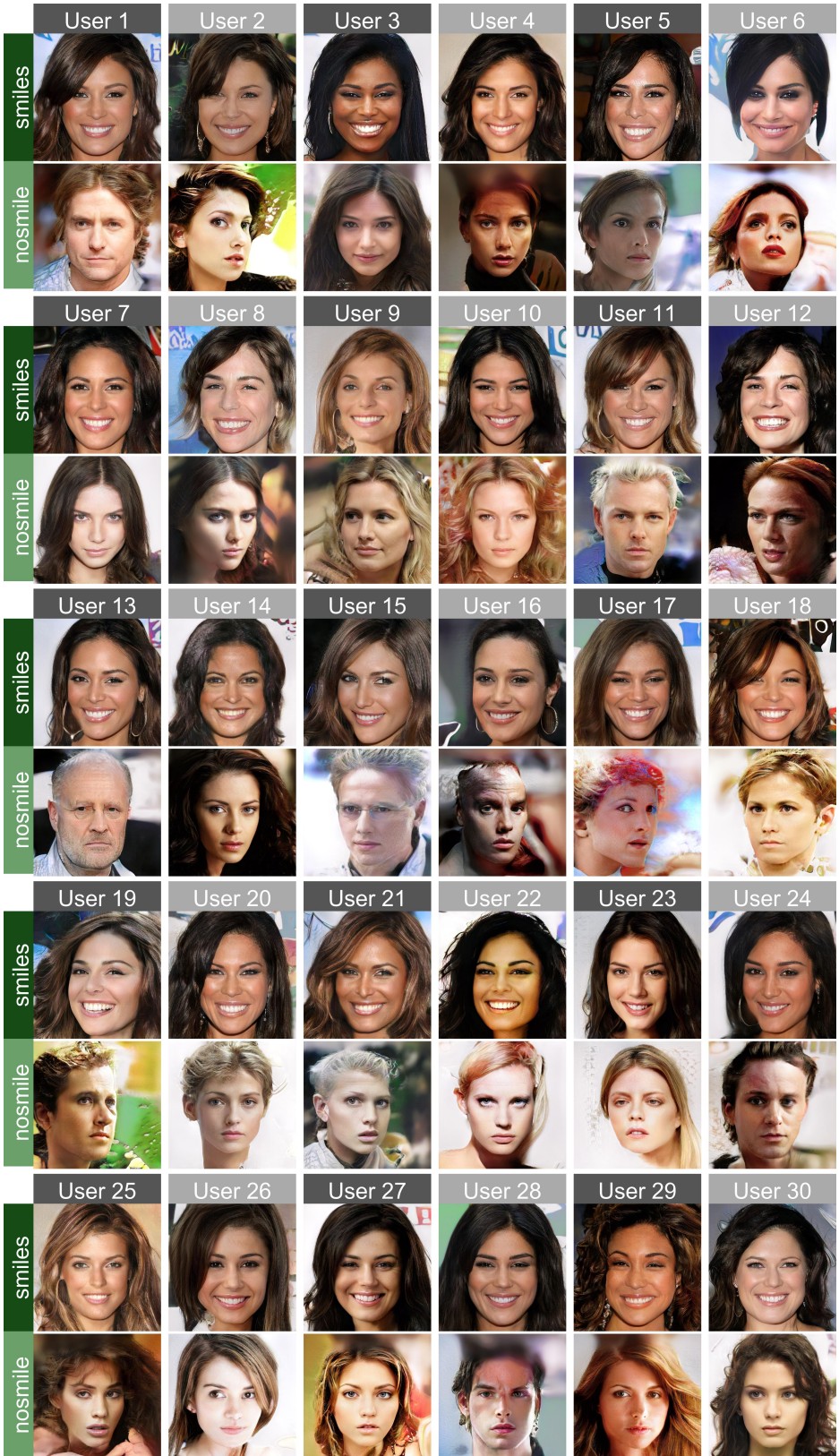

Figure A.12: Visualization of subsets of embeddings for each participant in task "smile" and "no smile", by mapping embeddings to the randomly sampled candidate set.

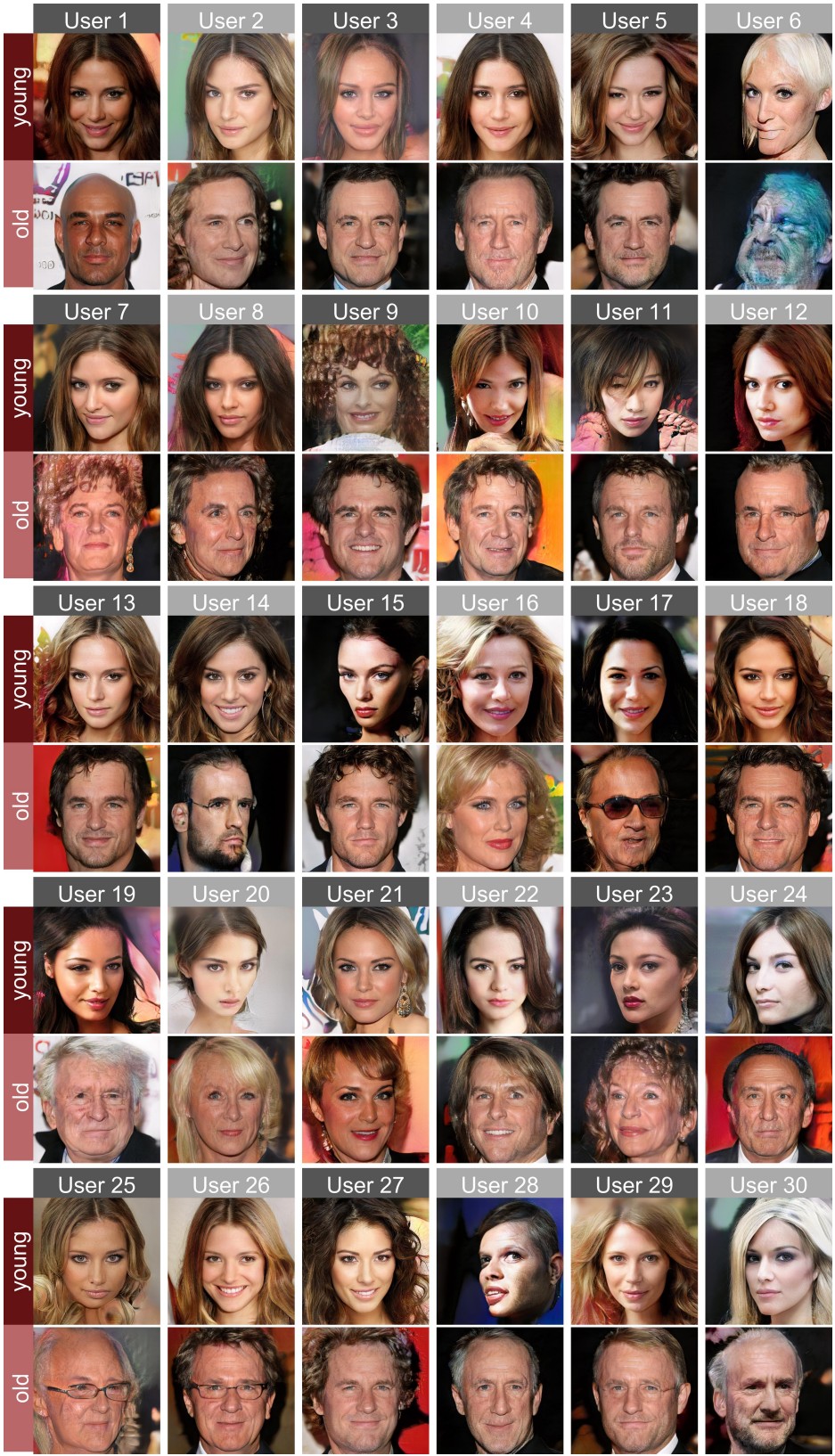

Figure A.13: Visualization of subsets of embeddings for each participant in task "young" and "old", by mapping embeddings to the randomly sampled candidate set.

