# OpenReview forum: "Cognition-Supervised Learning: Contrasting EEG Signals and Visual Stimuli For Saliency Detection"
_ICLR.cc/2024/Conference — ICLR 2024 Conference Withdrawn Submission_

### Official Review · Reviewer_h8j5 · 2023-10-19

**Soundness:** 2 fair
**Presentation:** 2 fair
**Contribution:** 2 fair
**Rating:** 3
**Confidence:** 3

**Summary:**

This paper uses contrastive learning for decoding EEG signals. The visual stimuli are annotations, the brain responses are the data, and both are matched using a contrastive loss instead of a classical supervised loss (e.g. least-squares). The authors empirically show that the learned representations cluster around annotation features (e.g. "young" or "old").

**Strengths:**

The paper applies contrastive learning to EEG signals and demonstrates evidence that the learned representations cluster around interesting features, including age group.

**Weaknesses:**

The writing style is at times over-emphatic compared to the actual contributions. For example, the abstract sets up the contribution as something completely novel and major ("remains paramount", "practically impossible", "novel paradigm", "the first of its kind"). However, in my understanding, this paper applies the existing framework of supervised contrastive learning [1] to EEG signals. Many relevant works are not cited: for example, consider [2] which also uses contrastive learning to learn representations from EEG signals and also demonstrates that the representations cluster following age group and other features.

Mapping the brain response to the stimulus is called "decoding" in the neuroscience literature [3]: it would be worth using this terminology.

Some citations could be added to support strong statements that begin a paragraph, e.g. "as human is known to respond strongly to facial stimuli" (missing citation) or "leverages a fundamental observation that the human brain response to differences in perception" (missing citation).

Some technical terms (e.g. "visual saliency" and "semantic saliency" in the abstract, "epochs" in the preprocessing paragraph) are used without being defined. Other terms ("target and non-target epochs") are also vague and undefined.


[1] Khosla et al. Supervised Contrastive Learning. NeurIPS 2020.

[2] Banville et al. Uncovering the structure of clinical EEG signals with self-supervised learning. Journal of Neural Engineering, 2021.

[3] King et al. Encoding and Decoding Neuronal Dynamics: Methodological Framework to Uncover the Algorithms of Cognition. Preprint, 2017.

**Questions:**

It might seem that the evaluation procedure is biased by design. For example, the "clusterability" of the learned representations around annotation features (e.g. young and old) is used to evaluate the learning procedure. Yet it is known that a contrastive loss performs clustering by design [1], so that "positive" pairs (here, a stimulus and corresponding brain response) are given "close" embeddings, and "negative" pairs (here, a stimulus and random brain response) are given "far" embeddings. So Table 3 and Figure 3 are entirely expected. Could the authors comment on that?

Some details are missing in the data preprocessing. For example, when "including a band-pass filter", could the authors specify the frequency-range in the main text?

[1] Wang et al. Understanding Contrastive Representation Learning through Alignment and Uniformity on the Hypersphere. ICML, 2020.

---

> ### Author Response · Authors · 2023-11-21
>
> 1. Regarding innovativeness, while we are not the first to apply self-supervision methods to EEG signals, our work is pioneering in applying these methods contrastively with visual stimuli. Prior research, such as sleep staging [1], has focused on using EEG signals alone for self-supervision. Our method's novelty lies in simultaneously utilizing both stimuli information and EEG signals, which are temporally aligned, a common practice in lab-collected EEG data and feasible in real scenarios with multiple sensors. We posit that using stimuli alongside EEG signals facilitates learning and identifying EEG structures, particularly in response to semantically salient stimuli, rather than deciphering these structures from scratch.
> Our approach differs from existing frameworks like SimCLR (unsupervised) and SupCon (supervised) that focus on a single modality (e.g., EEG signals) and typically require large datasets and batch sizes (4096 for SimCLR, 2048 for SupCon) for effective training. In contrast, EEG data collection is challenging, and our dataset comprises approximately 4400 data pairs for each task post-artifact removal, which is considerably smaller.
> We acknowledge that our initial manuscript may have overstated our contributions. We have revised the abstract, introduction, and conclusion to more accurately reflect our work's novelty and scope.
>
> [1] Jiang, Xue, et al. "Self-supervised contrastive learning for EEG-based sleep staging." 2021 International Joint Conference on Neural Networks (IJCNN). IEEE, 2021.
>
> [2] Chen, Ting, et al. "A simple framework for contrastive learning of visual representations." International conference on machine learning. PMLR, 2020.
>
> [3] Khosla, Prannay, et al. "Supervised contrastive learning." Advances in neural information processing systems 33 (2020): 18661-18673.
>
> 2. We appreciate the suggestion to use the term 'decoding'. We consciously chose not to use 'decoding' in our paper as our method focuses on learning an embedding from EEG signals that maps to only task-relevant parts of stimuli, which slightly diverges from traditional 'decoding' that maps EEG directly to stimuli. In our view, 'decoding' more accurately describes models that translate EEG signals into explicit labels, such as 'target' or 'non-target,' or identify specific stimuli, like a digit observed by a participant. Our approach, however, is a generalization of this task. It does not create a one-to-one mapping between EEG and stimuli but instead generates a versatile embedding applicable to various downstream tasks not limited to clustering, such as the conditional image generation discussed in Section 4.4 of our paper.
> To clarify our position, we have added a brief discussion on this distinction in the related work section of our revised manuscript.

---

> ### Author Response · Authors · 2023-11-21
>
> 3. Regarding the query on model performance and perceived evaluation bias, we contend that our results are not 'entirely expected' or trivial. The inherent noise in EEG signals makes even binary classification a significant challenge, even so in supervised BCI settings. In such supervised scenarios with complex visual stimuli, accuracies typically hover around 0.7 or lower in real-world applications, as seen in studies [1], [2], and [3]. As shown in Table 2 of our paper, the supervised EEGNet achieves a mean accuracy of only 0.699 across all tasks. Our model achieves comparable performance without relying on any external labels.
> +
> Considering the potential for signal distortion due to artifacts like eye blinks, the correlation of EEG signals to stimuli is not guaranteed, underscoring the difficulty of brain-computer interfacing tasks. To better illustrate this point, we have included additional discussions and relevant references in the related work section of our revised manuscript. Additionally, we have added tSNE plots in the Appendix A.5 of the revised manuscript to visually represent the embedding model's effectiveness before and after training.
>
> [1] Bagchi, Subhranil, and Deepti R. Bathula. "EEG-ConvTransformer for single-trial EEG-based visual stimulus classification." Pattern Recognition 129 (2022): 108757.
>
> [2] Lawhern, Vernon J., et al. "EEGNet: a compact convolutional neural network for EEG-based brain–computer interfaces." Journal of neural engineering 15.5 (2018): 056013.
>
> [3] Ahmed, Hamad, et al. "Object classification from randomized EEG trials." Proceedings of the IEEE/CVF Conference on Computer Vision and Pattern Recognition. 2021.
>
>
>
>
> 4. On the band passing details, it is 0.2 - 35 Hz, as briefly explained in Appendix A.1 data preprocessing section. This information is now added to the main text in section 3.1 in the revised manuscript. More details: the raw eeg data are processed with MNE (version 1.5.1), it is first reampled to 500Hz using mne.io.raw.resample(500), then filtered using mne.io.raw.filter(0.2, 35). These details will be added to Appendix A.1.

---

> > ### Comment · Reviewer_h8j5 · 2023-11-22
> > **Acknowledging author response**
> >
> > I thank the authors for their response which clarifies some of my concerns.
> >
> > **Expected results?** The contrastive training used by the authors --- by design --- pulls together EEG embeddings that are paired with similar stimuli, and pushes EEG embeddings that are paired with differing stimuli. In other words, the training clusters EEG embeddings based on the stimulus which acts as a label. So evaluating the embeddings by checking how well they predict (features of) the label that was used to cluster them seems biased at first glance. I would note that this concern seems to have been independently brought up by another reviewer (VsJo). Now, I understand the authors' point that their results are actually not so obvious. First, because the noise level in EEG data may be better handled by their contrastive training than by a fully supervised procedure where the EEG is used to predict the age / sex / hair color. If this is a claim the authors make, it should be clearer and substantiated. Second, because the EEG embeddings cluster based on one measure of similarity (scalar product) between stimuli used for training, but that does not necessarily imply that the EEG embeddings have to cluster based on another measure of similarity (features such as age / sex / hair color) between stimuli used for evaluation. This should be discussed much more clearly in order to guide the reader in understanding in what ways the authors' results could be construed as obvious but are maybe not so obvious.
> >
> > **Is it really self-supervised learning?**: the authors use the stimulus as a label during training, and then predict (features of) the stimulus during evaluation. It does raise the question whether this is actually supervised (not self-supervised) learning, which is a valid question brought up by Reviewer yZyL. I understand that there is a subtle point here, which is that that the "entire" stimulus is used for training, while evaluation consists in predicting a "latent class" of the stimulus (e.g. age / sex / hair color). So it is not exactly as if the same labels are used for training and evaluation. This should be further clarified in the main text as it is a point that was also brought up by another reviewer (yZyL).
> >
> > Overall, the authors' empirical results are interesting but more clarification is needed in the main text.

---

> > > ### Author Response · Authors · 2023-11-22
> > >
> > > Thank you for acknowledging our response.
> > >
> > > We affirm that our approach fits within the broader concept of self-supervised learning and multimodal learning. The potential confusion might stem from our model being trained with paired EEG-stimuli data that constits of two different modalities. However, it's crucial to note that during the training phase of the embedding model, the stimuli are not labeled. Our work build on top of the well-known self-supervision approach CLIP [1] where the idea is to find an representation from paired text and image data that aligns between two modalities, and in our case, the embedding model aligns representations from paired EEG and stimuli, all without labels. Our approach is also analogous to contrastive methods like ViLBERT (Vision & Language BERT) [2], which uses paired <text, image> data in its pre-training phase, while our model uses paired <EEG, image> data to learn the embeddings. And in [3], paired <video, audio> data is used for training. In all these related work, the learned representations are evaluted with downstream tasks, including but not limited to the linear classification tasks as we did in section 4.2. Unlike many self-supervised learning methods that focus on a single modality and rely on augmented views, our model contrasts EEG against unlabeled image stimuli, which comes effortlessly from the data collection steps. As discussed in section 2.1 of [1], the term "self-supervised learning" might be vague in the literature of multimodal learning. Analogous to CLIP which classifies itself as natural language supervision, we think the term cognition supervised learning is appropriate for our work. We added this discussion of the multimodal learning to the related work section of the revised manuscript.
> > >
> > >
> > > Considering the inherent noise in single-trial EEG signals, even supervised binary classification is challenging. For instance, accuracies is 0.78 in [4] for classifying human faces against objects, around 0.7 in [5] for within-subject and around 0.4 for cross-subject, 0.708 in [6] and below 0.6 for text stimuli in [7]. Our primary contribution lies not in surpassing fully supervised methods in EEG-based classification, but in developing a pre-training method to learn versatile EEG representations. These representations can be applied to a range of downstream tasks, as demonstrated in our experiments with classification and conditional generation. This versatility is akin to the work in [3], where a representation learned from paired audio-visual data is later applied to tasks like sound localization or audio-visual source separation. Our method’s potential extends beyond the tasks explored in our study, opening new avenues for research that can learn representations that match human experiences as mesured directly from human cognition.
> > >
> > > [1] Radford, Alec, et al. "Learning transferable visual models from natural language supervision." International conference on machine learning. PMLR, 2021.
> > >
> > > [2] Lu, Jiasen, et al. "Vilbert: Pretraining task-agnostic visiolinguistic representations for vision-and-language tasks." Advances in neural information processing systems 32 (2019).
> > >
> > > [3] Owens, Andrew, and Alexei A. Efros. "Audio-visual scene analysis with self-supervised multisensory features." Proceedings of the European conference on computer vision (ECCV). 2018.
> > >
> > > [4] Bagchi, Subhranil, and Deepti R. Bathula. "EEG-ConvTransformer for single-trial EEG-based visual stimulus classification." Pattern Recognition 129 (2022): 108757.
> > >
> > > [5] Lawhern, Vernon J., et al. "EEGNet: a compact convolutional neural network for EEG-based brain–computer interfaces." Journal of neural engineering 15.5 (2018): 056013.
> > >
> > > [6] Ahmed, Hamad, et al. "Object classification from randomized EEG trials." Proceedings of the IEEE/CVF Conference on Computer Vision and Pattern Recognition. 2021.
> > >
> > > [7] Eugster, Manuel JA, et al. "Predicting term-relevance from brain signals." Proceedings of the 37th international ACM SIGIR conference on Research & development in information retrieval. 2014.

---

### Official Review · Reviewer_VsJo · 2023-10-31

**Soundness:** 3 good
**Presentation:** 3 good
**Contribution:** 3 good
**Rating:** 6
**Confidence:** 5

**Summary:**

The paper propose an interesting idea called "cognition-supervised learning". The method uses EEG signals as direct supervisory information. The approach uses EEG to contrastively train models to detect visual saliency without the need for manual annotations. The paper then applies the learned representations for several downstream tasks (classification, clustering, and image generation) showing competitive performance compared to models trained with manually labeled datasets.

**Strengths:**

The paper provides a promising new direction for research where brain signals such as EEG can be utilized directly for supervising deep learning models. The paper is well-written.

**Weaknesses:**

I ask the authors to respond to the following weaknesses/questions:

- There has been prior works on image reconstruction from EEG, e.g.
"NeuroGAN: image reconstruction from EEG signals via an attention-based GAN"
"EEG2IMAGE: Image Reconstruction from EEG Brain Signals"
"Photorealistic Reconstruction of Visual Texture From EEG Signals"
"Visual Saliency and Image Reconstruction from EEG Signals via an Effective Geometric Deep Network-Based Generative Adversarial Network", and others.
Given that the proposed paper seems to perform the opposite direction of these works, they could be discussed. Moreover, simple reverse versions of these can be developed to use as baselines.

- It would be valuable to see the impact of the CLIP loss by exploring and comparing other ones as well.

- There's a small error in Fig 1, "Find-tuning" --> Fine-tuning

- While the RQs are very interesting, I wonder whether the outcome is completely expected. Given that we can map EEG to certain classes/actions/objects (e.g., affect classes, objects, directions, etc.), doesn't the outcome become expected? In other words, the method is mapping two high-dimensional data points onto a common lower dimensional feature space, where the locality of the image embeddings is used as the supervisory signal. Can the authors please (a) provide a discussion on this, (b) show the common embedding space (using tSNE, UMAP, etc) to extract some more insights about the reason behind why the method works. In general, I found the "analysis" part of the paper a bit weak. Further experiments to analyze the method, its components, and embedding space, are recommended.

**Questions:**

Please see my comments under weaknesses.

---

> ### Author Response · Authors · 2023-11-21
>
> Regarding prior works on EEG-based image reconstruction, our introduction section discusses a series of earlier studies suffering from confounded EEG data due to specific experimental block designs. This includes the EEG-GAN approach [1], [2], Thoughtviz [3], Brain2image [4], EEG-ChannelNet[5], and subsequent research such as EEG2IMAGE [6], DM-RE2I [7], NeuroGAN [8], and others [9]. Many of these studies use the same datasets; for instance, NeuroGAN uses the same dataset from Thoughtviz, and [9] employs the EEG-ImageNet dataset from [5].
>
> However, subsequent analyses [10], [11], [12] have identified a critical flaw in these approaches: the block design in data collection introduces temporal correlations between the stimulus class and the experiment's duration due to the same presentation order of stimulus class. Replication attempts have suggested that models were learning to condition on this temporal correlation rather than stimulus-related brain activity.
>
> On the other hand, there are studies that utilize EEG signals from multiple trials to overcome the low signal-to-noise ratio inherent in EEG, like [13], which employs 24 repetitions of each stimulus and averages the visual evoked potentials for texture reconstruction. While these approaches have shown promising results, single-trial EEG-based high-resolution image reconstruction remains largely unexplored, to the best of our knowledge.
>
> We have appended this discussion to the related work section of our revised manuscript.
>
> [1] Palazzo, Simone, et al. "Generative adversarial networks conditioned by brain signals." Proceedings of the IEEE international conference on computer vision. 2017.
>
> [2] Spampinato, Concetto, et al. "Deep learning human mind for automated visual classification." Proceedings of the IEEE conference on computer vision and pattern recognition. 2017.
>
> [3] Tirupattur, Praveen, et al. "Thoughtviz: Visualizing human thoughts using generative adversarial network." Proceedings of the 26th ACM international conference on Multimedia. 2018.
>
> [4] Kavasidis, Isaak, et al. "Brain2image: Converting brain signals into images." Proceedings of the 25th ACM international conference on Multimedia. 2017.
>
> [5] Palazzo, Simone, et al. "Decoding brain representations by multimodal learning of neural activity and visual features." IEEE Transactions on Pattern Analysis and Machine Intelligence 43.11 (2020): 3833-3849.
>
> [6] Singh, Prajwal, et al. "EEG2IMAGE: Image reconstruction from EEG brain signals." ICASSP 2023-2023 IEEE International Conference on Acoustics, Speech and Signal Processing (ICASSP). IEEE, 2023.
>
> [7] Zeng, Hong, et al. "DM-RE2I: A framework based on diffusion model for the reconstruction from EEG to image." Biomedical Signal Processing and Control 86 (2023): 105125.
>
> [8] Mishra, Rahul, et al. "NeuroGAN: image reconstruction from EEG signals via an attention-based GAN." Neural Computing and Applications 35.12 (2023): 9181-9192.
>
> [9] Khaleghi, Nastaran, et al. "Visual saliency and image reconstruction from EEG signals via an effective geometric deep network-based generative adversarial network." Electronics 11.21 (2022): 3637.
>
> [10] Li, Ren, et al. "The perils and pitfalls of block design for EEG classification experiments." IEEE Transactions on Pattern Analysis and Machine Intelligence 43.1 (2020): 316-333.
>
> [11] Ahmed, Hamad, et al. "Object classification from randomized EEG trials." Proceedings of the IEEE/CVF Conference on Computer Vision and Pattern Recognition. 2021.
>
> [12] Ahmed, Hamad, et al. "Confounds in the data—Comments on “Decoding brain representations by multimodal learning of neural activity and visual features”." IEEE transactions on pattern analysis and machine intelligence 44.12 (2021): 9217-9220.
>
> [13] Wakita, Suguru, Taiki Orima, and Isamu Motoyoshi. "Photorealistic reconstruction of visual texture from EEG signals." Frontiers in Computational Neuroscience 15 (2021): 754587.

---

> ### Author Response · Authors · 2023-11-21
>
> We think that our results are not entirely expected. Indeed, our embedding model maps high-dimensional EEG data to a lower-dimensional embedding, preserving relevant structures with the aid of paired stimuli data. However, this mapping is not trivial for two key reasons.
>
> Firstly, the inherent noise in single-trial EEG signals makes even supervised binary classification a challenging task, often resulting in accuracies around 0.7. For instance, accuracies is 0.78 in [1] for classifying human faces against objects, around 0.7 in [2] for within-subject and around 0.4 for cross-subject,  0.708 in [3] and below 0.6 for text stimuli in [4]. This variability underscores that mapping EEG signals to a common feature space is not straightforward.
>
> Secondly, while we use the locality of image embeddings to reduce the dimensions of EEG signals, these image embeddings alone are insufficient to discern semantic saliency. Section 4.1 and Table 1 demonstrate that clusters based on image embeddings alone do not align with the semantic saliency of our tasks. However, when clustering the EEG embeddings in the common feature space, there is a notable alignment with the target semantic feature. Furthermore, a PCA on these learned embeddings reveals that the most significant component correlates with the task's targeted semantic feature. We acknowledge the value of a deeper exploration into the learned EEG embeddings and their properties beyond mere correlation with task semantics. To this end, we have added new tSNE plots of both raw EEG signals and the learned embeddings in Appendix A.4 of our revised manuscript.
>
> [1] Bagchi, Subhranil, and Deepti R. Bathula. "EEG-ConvTransformer for single-trial EEG-based visual stimulus classification." Pattern Recognition 129 (2022): 108757.
>
> [2] Lawhern, Vernon J., et al. "EEGNet: a compact convolutional neural network for EEG-based brain–computer interfaces." Journal of neural engineering 15.5 (2018): 056013.
>
> [3] Ahmed, Hamad, et al. "Object classification from randomized EEG trials." Proceedings of the IEEE/CVF Conference on Computer Vision and Pattern Recognition. 2021.
>
> [4] Eugster, Manuel JA, et al. "Predicting term-relevance from brain signals." Proceedings of the 37th international ACM SIGIR conference on Research & development in information retrieval. 2014.
>
> * Thank you for highlighting the typo in Figure 1. We have now corrected it in the revised manuscript.

---

### Official Review · Reviewer_4tkM · 2023-11-02

**Soundness:** 3 good
**Presentation:** 3 good
**Contribution:** 2 fair
**Rating:** 6
**Confidence:** 4

**Summary:**

This paper presents cognition-supervised learning using EEG data to contrastively train models for visual saliency without manual annotations, which generates competitive performance on subsequent tasks and opens the way for future research on human cognitive system-guided supervision. for computer vision and machine learning.

**Strengths:**

The paper introduces a original and innovative concept, "cognition-supervised learning," which leverages human brain signals (EEG data) as direct supervisory signals for training machine learning models.

The paper explores the concept of cognition-supervised learning and provides a well-structured experimental framework to validate its effectiveness.

The paper is well written and clearly communicates its key ideas, methodology, and results. The introduction clearly states the context and motivation for the research, and the research questions are clearly described.

The contribution with an open EEG dataset in this paper is a great contribution, as it not only encourages transparency and reproducibility, but also promotes collaboration and facilitates further exploration of cognition-supervised learning, encouraging growth and progress of the research field.

The paper provides a detailed description of the dataset used, along with a detailed explanation of the processing of the data involved in the study. This strengthens the methodological quality of the research.

The paper maintain a level of quality in its experiments. The use of well-established techniques like unsupervised clustering and linear classifiers ensures the reliability of the evaluation. In addition, the incorporation of a qualitative evaluation using generative adversarial networks adds a layer of quality.

To sum up, the use of EEG data for supervision in machine learning is a novel and interesting direction that could have positive implications for the field.

**Weaknesses:**

The paper discusses results in the context of facial images, but it is not clear how well the proposed approach generalizes to other types of images or domains. It would be important to conduct experiments with a larger range of datasets to demonstrate the robustness of the cognition-supervised learning approach.

The paper mentions that it opens avenues for human-in-the-loop systems, but it would be valuable to provide insights into potential future research directions and how this work could be extended and applied in various domains (e.g. NLP)

In section 3.1 the authors mention that "The images were manually screened to ensure realism and the absence of visual artifacts", but I think a little more detail should be given regarding this.

At the beginning of section 3.2, it would be desirable for authors to include relevant references to support these claims, thus facilitating a deeper exploration of their claims in the article.

In section 4.2, it would be good for the authors to compare the results with more control models (e.g. EEGNet Fusion, MI-EEGNet, etc.) in order to have stronger results. Furthermore, the difference between EEGNet and contrastive embedding (Mean column) is very small (0.699 ± 0.037 vs 0.704 ± 0.046), which even falls within the standard deviation.

As mentioned in the limitations, the present work only applies to two classes; it would be of great interest to expand it to more classes, since many problems are of this nature.

**Questions:**

Do the authors believe that the present work could be expanded to more classes and not just two?

Do the authors plan to expand the work with other types of images or in another domain (e.g. NLP)?

---

> ### Author Response · Authors · 2023-11-21
>
> 1. Regarding the generalizability of our approach and the use of facial images: We chose facial images due to their homogeneity in features such as color, shape, and geometry, which minimizes confounding factors in EEG experiments. For instance, using an image with a red sports car might evoke brain responses related to the color red or interest in cars, rather than the intended semantic features. To ensure the validity of our dataset and avoid such biases, we needed stimuli that varied semantically while being invariant to other confounding attributes.
> +
> Artificial facial images were specifically selected to further reduce confounding effects. Recognizing a celebrity could trigger brain responses associated with such recognition effect rather than the semantic saliency of facial features, which was our experimental focus.
> +
> As for image selection and processing, all 70,000 images were manually screened by several researchers. The criterion was to exclude images with visual artifacts, such as distorted faces or other clear signs of an artificial image, to prevent brain responses from being influenced by artifact recognition rather than semantic saliency. The set of 70,000 images were the starting point for sampling the stimuli, which was done in the grouping phase. We have extended appendix section A.1  of our revised manuscript to further clarify this.
>
> 2. Regarding the first question on the expansion of our work beyond binary tasks: Yes, we believe that our method for learning embeddings is indeed applicable to non-binary tasks. Our model is designed to learn a versatile representation of EEG responses to a variety of stimuli, not restricted to binary classes. However, expanding our approach to non-binary tasks would require careful consideration of experimental design, from stimuli selection to data preprocessing, to ensure a strong and meaningful correlation between EEG signals and the stimuli.
>
> 3. Regarding the second question on the potential application of our work with different types of stimuli: We see significant potential for applying our approach to stimuli beyond artificially generated facial images, primarily because it obviates the need for manual annotation of these stimuli. For various groups of images, one could use image encoders instead of latent codes as the representation of the stimuli, where the encoder can be trained on large image datasets. Similarly, it is feasible to employ stimuli in other modalities, such as text, audio, and video, leveraging existing large models that learn embeddings for these formats.
>
> 4. Regarding the results in Section 4.2 and Table 2, and the relatively modest performance difference between EEGNet and our method, we have expanded the interpretation in the results section to highlight a key distinction: our model is trained without labels. In contrast, EEGNet and its variants, such as EEGNet Fusion and MI-EEGNet, are supervised models trained with ground truth labels. In our linear evaluation, even though a supervised linear classifier is trained on the frozen embeddings from our model, its performance is naturally expected to be somewhat inferior to models trained entirely under supervised conditions. This is consistent with findings in other studies, such as simCLR [1], where the ImageNet Top-1 accuracy of a fully supervised ResNet-50 is above 75%, compared to 69.3% for a linear classifier trained on top of simCLR embeddings, despite using the same architecture.
>
> [1] Chen, Ting, et al. "A simple framework for contrastive learning of visual representations." International conference on machine learning. PMLR, 2020.

---

### Official Review · Reviewer_yZyL · 2023-11-09

**Soundness:** 1 poor
**Presentation:** 2 fair
**Contribution:** 1 poor
**Rating:** 3
**Confidence:** 5

**Summary:**

The proposed approach for self-supervised machine learning, which involves using human brain signals as direct supervisory signals, raises questions. The authors suggest a paradigm of cognition-supervised learning achieved through leveraging EEG data to receive solid binary labels. However, using BCI-style responses for such self-supervised learning could be more ethically questionable and logically messy, as it depends on human involvement and may not have practical applications. Moreover, the study assumes involuntary or automatic human responses, which raises ethical concerns.

---
POST-AUTHOR FEEDBACK:
---
In response to helpful feedback, the reviewer upgraded their decision, however, the manuscript remains unsuitable for publication due to unresolved ethical concerns and questionable self-supervision application.

**Strengths:**

It is quite challenging to identify any strengths in the manuscript. The claims of self-supervision using "hard P300" labels provided by humans who were forced to label pictures in visually tiring conditions (all BCI P300-style experiments are actually exhausting) are not well-explained and validated. It is difficult to identify the novelty of the model, and the final results are not impressive.

**Weaknesses:**

It is not clear how the concept adheres to the self-supervision philosophy, as efficient labels like those of P300 are actually human-generated labels. Self-supervised learning is typically considered a form of unsupervised learning where the model generates its own labels or annotations from the input data, without the need for external human-labeled data. This involves creating a pretext or auxiliary task that does not require human annotations. The model is trained to predict some aspect of the input data based on other parts of the same data. However, P300 responses use human-generated labels in an ethically questionable setup.

**Questions:**

The standard 10-20 EEG electrode placement system consists of 21 electrodes. However, in this case, 32 electrodes were used. I am curious about the placement of these additional electrodes on the head and the reason for using 32 instead of the usual single Pz electrode. Moreover, I am concerned about the ethical implications of violating the Declaration of Helsinki by misusing human subjects.

**Details Of Ethics Concerns:**

The concept of using individuals as "involuntary labelers" and potentially compromising privacy by manipulating difficult-to-control automatic reactions is ethically controversial. This questionable technology, which employs human faces, could infringe upon the privacy of subjects' sexual preferences and other personal information. The study's logic is also flawed, as it does not appear to involve self-supervised training, rather, it utilizes a convoluted approach to supervised training which misleads and violates the EEG experiment participants' ethical rights.

---

> ### Author Response · Authors · 2023-11-21
>
> Regarding the definition of self-supervision, our model is trained without explicit labels; it uses only EEG data and images during training. Labels are employed solely for performance evaluation. Similar EEG-based self-supervised methods have been applied to sleep stage classification [1], emotion recognition [2] and pathology screening [3]. Our approach is similar to prior contrastive learning methods SimCLR [4] and supervised contrastive learning SupCon [5], where a contrastive loss is used in the self-supervised setting. However, our approach differs from these existing frameworks that focus on a single modality (e.g., EEG signals) and typically require large datasets and batch sizes (4096 for SimCLR, 2048 for SupCon) for effective training. In contrast, EEG data collection is challenging, and our dataset comprises approximately 4400 data pairs for each task post-artifact removal, which is considerably smaller.
>
> While EEG signals are indeed human-generated inputs, with participants actively engaged in the task, the mixed stimuli images which include both target and non-target semantics, make the task of decoding EEG signals to a binary label particularly challenging, more so than in typical supervised settings where labels are provided. In such supervised scenarios with complex visual stimuli, especially in true single-trial binary classification tasks where repetition of stimuli classes is not allowed, the typical accuracy often hovers around 0.7, and can be even lower in real-world applications, for example 0.78 in [6] for classifying human faces against objects, around 0.7 in [7] for within-subject and around 0.4 for cross-subject,  0.708 in [8] and below 0.6 for text stimuli in [9]. Our model achieves comparable performance without relying on any external labels.
> To enhance the clarity, we added the above discussion and references  to the introduction and related work section of our revised manuscript.
>
> [1] Jiang, Xue, et al. "Self-supervised contrastive learning for EEG-based sleep staging." 2021 International Joint Conference on Neural Networks (IJCNN). IEEE, 2021.
>
> [2] Mohsenvand, Mostafa Neo, Mohammad Rasool Izadi, and Pattie Maes. "Contrastive representation learning for electroencephalogram classification." Machine Learning for Health. PMLR, 2020.
>
> [3] Banville, Hubert, et al. "Uncovering the structure of clinical EEG signals with self-supervised learning." Journal of Neural Engineering 18.4 (2021): 046020.
>
> [4] Chen, Ting, et al. "A simple framework for contrastive learning of visual representations." International conference on machine learning. PMLR, 2020.
>
> [5] Khosla, Prannay, et al. "Supervised contrastive learning." Advances in neural information processing systems 33 (2020): 18661-18673.
>
> [6] Bagchi, Subhranil, and Deepti R. Bathula. "EEG-ConvTransformer for single-trial EEG-based visual stimulus classification." Pattern Recognition 129 (2022): 108757.
>
> [7] Lawhern, Vernon J., et al. "EEGNet: a compact convolutional neural network for EEG-based brain–computer interfaces." Journal of neural engineering 15.5 (2018): 056013.
>
> [8] Ahmed, Hamad, et al. "Object classification from randomized EEG trials." Proceedings of the IEEE/CVF Conference on Computer Vision and Pattern Recognition. 2021.
>
> [9] Eugster, Manuel JA, et al. "Predicting term-relevance from brain signals." Proceedings of the 37th international ACM SIGIR conference on Research & development in information retrieval. 2014.

---

> ### Author Response · Authors · 2023-11-21
>
> Regarding the number of electrodes, their placement, and montage: The 10-20 electrode placement system, introduced in 1958, served as the standard relative head surface positioning method for many years. However, with the advent of multichannel EEG hardware systems, there has been a shift towards higher EEG electrode density. This shift led to the development and adoption of the 10-10 and later the 10-5 systems as new standards by the American Clinical Neurophysiology Society [Ref:soc1] and the International Federation of Clinical Neurophysiology [Ref:fed1]. Modern EEG equipment typically use 32, 64, or upto 128 channels. We utilize the standard 32 electrode set and placement.
> Specifically, we used 32 equidistant electrodes situated at FP1, FP2, F7, F3, Fz, F4, F8, FC5, FC1, FC2, FC6, T7, C3, Cz, C4, T8, TP9, CP5, CP1, CP2, CP6, TP10, P7, P3, Pz, P4, P8, PO9, O1, O2, PO10, Iz, within the 10% system, as explained in the appendix A.1 of our revised manuscript. We also added a plot of the electrodes placement in the Appendix section A.1.
>
> [Ref:soc1] American Electroencephalographic Society. Guideline Thirteen: guidelines for standard electrode position nomenclature. J.Clin.Neurophysiol 1994;11:111–113.
>
> [Ref:fed2] Nuwer M R ,Comi G ,Emerson R,et.al. IFCN standards for digital recording clinical EEG.Electroencephalogr Clin Neurophysiol 1998;106:259–261.
>
> Regarding ethical concerns, we have worked with the highest ethical standards set for neuroscientific research and research involving human participants. Our experiments are accepted by the ethical review board of BLINDED FOR REVIEW with a full ethical evaluation of the experiments, intention of use and the handling and management of data. Our experiments fully comply with general ethical guidelines of the world medical association (declaration of Helsinki). The data acquisition experiments have been run by experienced neuroscience researchers. We have detailed these ethical considerations extensively in our paper in Ethics Statement. Additionally, our approach to decoding human brain responses for research is in line with practices in the field, as evidenced by related studies published in journals like Nature Neuroscience [nat1, nat2, nat3, nat4]. We have not employed any form of 'involuntary labeling' in our research, and we are puzzled by the reviewer’s interpretation in this regard. We respectfully request a careful re-examination of the ethical discussions already present in our manuscript and expect a fair assessment of our work in the ICLR review process.
>
> [nat1] Défossez, Alexandre, et al. "Decoding speech perception from non-invasive brain recordings." Nature Machine Intelligence (2023): 1-11.
> [nat2] Tang, Jerry, et al. "Semantic reconstruction of continuous language from non-invasive brain recordings." Nature Neuroscience (2023): 1-9.
> [nat3] Wolff, Michael J., et al. "Dynamic hidden states underlying working-memory-guided behavior." Nature neuroscience 20.6 (2017): 864-871.
> [nat4] Van Ede, Freek, et al. "Concurrent visual and motor selection during visual working memory guided action." Nature Neuroscience 22.3 (2019): 477-483.
>
>
> In summary, we are sorry to state this, but this review is not fair and the reviewer does not seem to be knowledgeable about the EEG specifics  nor have carefully read the ethical protocol that we have described in the paper, or is not familiar with this type of research. We hope that the reviewer and the area chair can reconsider this review and reflect it according to our responses.

---

> ### Comment · Reviewer_yZyL · 2023-11-22
> **No change in the reviewer decision**
>
> The decision of the reviewer remained unchanged, even after the authors provided feedback. Although the authors responded in detail, they were unable to provide sufficient justification for their use of self-supervision. The use of the human brain as an external "labeler" by the authors raised ethical concerns that were not adequately addressed, despite obtaining a successful local ethical review. The reviewer refrained from commenting on the authors' evaluations of the EEG experience. Additionally, the reviewer carefully considered the remaining reviews with feedback, but they did not change the original decision.

---

> > ### Author Response · Authors · 2023-11-22
> > **Request for area chair to take an opinion**
> >
> > I am writing as the senior author of the submitted manuscript. There seems to be a disagreement between the reviewer and the authors on ethical concerns of using human subjects as "involuntary labelers". We have not conducted any research that would propose involuntary labeling. Instead, we follow strict ethical protocols set out by WHO and our university ethical board from which we have obtained permission to conduct this research. It is important to recognize that the type of research where human brain responses are used as a source for machine learning input is very common and extensively published aslo in neuroscience venues, as we have also pointed out in our previous response.
> >
> > What comes to the description of self-supervision, the first author has already responded to such comments. Please see the other responses in this thread for clarity.
> >
> > I find the scores given by the reviewer and the grounds to criticize the manuscript unjustified. Therefore, I would ask the area chair to carefully consider this disagreement to ensure a fair review process.

---

> > > ### Comment · Reviewer_yZyL · 2023-11-22
> > > **Further clarification**
> > >
> > > Thank you very much for getting back to me so quickly from the senior co-author of the submission. The reviewer agrees that the area chair could provide helpful comments. It is possible to counteract a single negative review in such cases.
> > >
> > > To clarify, the reviewer does not question the ethics of the particular experiment reported in the submission. The classical P300 experiment, which involved mentally discriminating targets from non-targets, was probably completed according to worldwide ethical standards. The issue lies with the broader moral implications of such experiments and applications, where humans "could be used" as more or less cooperating entities, potentially in a setup similar to the one in the movie THE MATRIX. As ICLR is a highly influential conference, such publications could cause more harm to the field despite already published papers on similar subjects. Unfortunately, the humble reviewer is not in a position to comment on them. If the authors could persuade the reviewers that such technology could bring more good than harm, it would convince the reviewer yZyL to reconsider the evaluation. The choice of P300, probably the best in the BCI/EEG field, was brilliant. However, the solid ERP somehow killed the concept of self-supervision.
> > >
> > > This issue of "self-supervision" has also caused concern in the reviewer's h8j5 recent comment. Therefore, this is another point where the area chair could help decide.

---

> > > > ### Author Response · Authors · 2023-11-22
> > > > **On hypothetical misuse of the proposed technology**
> > > >
> > > > Thank you for your response. We sincerely wish that advancements in technology are employed for positive purposes rather than negative ones. There are certainly many potential applications of our approach for the benefit of humanity. For example, but not limited to, 1) assisting people with disabilities (such as locked-in patients or patients with limited motor abilities) to express their preferences and create new information using BCI. 2) Assist neuroscience research by providing a new methodology that is not dependent on labeled data and 3) that could be used with natural stimuli. Here, we mean learning how humans respond to certain information without making any pre-assumptions on the labels associated with that information. This is one of the key benefits of the reported research -- although we use labels in our evaluation to empirically validate that the method works. 4) Clinical research methodologies for neuropsychology, and 5) new types of human-computer interaction. For example, in virtual reality stimulus and physiological data can be simultaneously obtained, but labels on how humans react to stimuli are not available. We will add a statement of potential applications to the manuscript and remind that we already have an ethical statement in the manuscript, where we discuss potential impact.

---

### Meta-Review · Area_Chair_nNUz · 2023-12-10

**Metareview:**

This work proposes to use a contrastive loss (CLIP) between visual stimuli and EEG in order to supervise the representation learning. It is argued that this allows to orient the representation towards salient information, which is a task that the brain is very good at. The approach is referred to as "cognition-supervised learning". To deal with inter-subject variability a subject embedding is employed.

The paper comes also with a new dataset of 30 subjects.

The paper has been judged overall well written without any experimental flow. The idea of "cognition-supervised learning"
was also judged original.

The work is however challenged on its practical impact and its actual novelty in terms of machine learning method.
For this reason and given the strong expectations of a top conference like ICLR, I cannot endorse this work for publication
this year.

**Justification For Why Not Higher Score:**

Limited ML contribution and limited practical impact.

**Justification For Why Not Lower Score:**

Well written and experiments without identified flaw.

---

### Decision · Program_Chairs · 2024-01-16

Reject